Representing winter wheat in the Community Land Model (version 4.5)
Yaqiong Lu[1,2*], Ian N. Williams[1], Justin E. Bagley[1], Margaret S. Torn[1,3], Lara M.
Kueppers[1,3]
[1]*Climate and Ecosystem Sciences Division, Lawrence Berkeley National Laboratory*
[2]*Climate and Global Dynamics Laboratory, National Center for Atmospheric Research*
[3]*Energy and Resources Group, University of California, Berkeley*
*Corresponding author: Yaqiong Lu, yaqiong@ucar.edu, 303-497-1389, 1850 Table
Mesa Drive, Boulder, CO 80305
Abstract
Winter wheat is a staple crop for global food security, and is the dominant vegetation
cover for a significant fraction of Earth's croplands. As such, it plays an important role in
carbon cycling and land-atmosphere interactions in these key regions. Accurate
simulation of winter wheat growth is not only crucial for future yield prediction under
changing climate, but also for well predicting the energy and water cycles for winter
wheat dominated regions. We modified the winter wheat model in the Community Land
Model (CLM) to better simulate winter wheat leaf area index, latent heat flux, net
ecosystem exchange of $CO_2$, and grain yield. These included schemes to represent
vernalization, as well as frost tolerance and damage. We calibrated three key parameters
(minimum planting temperature, maximum crop growth days, and initial value of leaf
carbon allocation coefficient) and modified the grain carbon allocation algorithm for
simulations at the U.S. Southern Great Plains ARM site (US-ARM), and validated the
model performance at eight additional sites across North America. We found that the new
winter wheat model improved the prediction of monthly variation in leaf area index,
reduced latent heat flux and net ecosystem exchange RMSE by 41% and 35% during the
spring growing season. The model accurately simulated the interannual variation in yield
at the US-ARM site, but underestimated yield at sites and in regions (Northwestern and
Southeastern US) with historically greater yields by 35%.
Introduction
Wheat is a widely grown temperate cereal (Shewry, 2009), ranked fourth among
commodity crops with a global production of 711 million tonnes, and encompasses
13.3% of global permanent cropland as of 2013 (http://faostat3.fao.org/home/E). Wheat
provides one-fifth of the total caloric input of the world's population (Curtis et al., 2002),
and therefore plays an important role in global food security (Chakraborty and Newton,
2011; Vermeulen et al., 2012). In many regions, such as the United States, winter wheat
(*Triticum aestivum*) is the dominant wheat cultivar accounting for 74% of the total U.S.
wheat production, based on data from the National Agricultural Statistics Service of the
U.S. Department of Agriculture in 2013 (http://www.nass.usda.gov).
Winter wheat, which is planted in fall and harvested in early summer, has a different
growth cycle and responds to environmental stresses differently from summer crops.
Winter wheat may suffer less from summer drought but is subject to winter damage due
to exposure to low temperatures and frequent freeze-thaw cycles (Vico et al., 2014).
There are two important over-winter survival mechanisms for winter wheat: vernalization
and cold tolerance. Vernalization is the process whereby winter wheat is exposed to a
period of non-lethal low temperature required to fully enter the flowering stage and to
produce grain in spring (Chouard, 1960). Additionally, winter wheat acclimates to low
temperature giving it the capability to survive cold temperatures. Both of these processes
– vernalization and cold tolerance - are cumulative processes and have similar optimum
temperature ranges. When the temperature is outside of the optimum range, the processes
can be stopped, reversed, and restarted (Fowler et al., 1999). Damage can occur when
temperatures are lower than the accumulated cold tolerance (reviewed by Barlow et al.,
(2015)). Cold-induced damage has been observed to persist through the remainder of the
growing season, and its impact on yield is greater than on growth. Effectively
representing these processes in crop models could improve understanding of the
uncertainty in the future crop yield projections.
Winter wheat also plays an important role in land-atmosphere interactions through effects
on energy, water, and carbon fluxes. Winter wheat cropland has much less soil carbon
loss compared to maize cropland averaged across several sites (Ceschia et al., 2010), and
could either be a carbon sink (Waldo et al., 2016) or source (Anthoni et al., 2004),
depending on the year and the location.  The earlier growing season can influence surface
fluxes of water, energy, and momentum, and hence regional climate (Riley et al., 2009).
This land surface influence is particularly strong in the U.S. Southern Great Plains, where
winter wheat is a dominant land-cover type. For example, statistical analyses indicated
cooler and moister near-surface air over Oklahoma's winter wheat belt from November to
April compared to adjacent grassland, due to the influence of winter wheat (McPherson et
al., 2004). This influence highlights the importance of adequately representing winter
wheat in land surface models used for climate projections, in order to assess both the
impact of climate change on agriculture and agriculture's influence on regional climate.
The agricultural research community developed several winter wheat models during the
1980s, such as the Agricultural Research Council winter wheat model (ARCWHEAT)
(Porter, 1984; Weir et al., 1984) and the Crop Estimation through Resource and
Environment Synthesis winter wheat model (CERES-wheat) (Ritchie and Otter, 1985).
These models were designed to simulate winter wheat growth at the farm level and have
well-defined winter wheat growth phenology, which is a function of thermal time and day
length that are adjusted by vernalization and a photoperiod factor. Photosynthesis and
respiration processes determine the dry matter for partitioning among roots, shoots,
leaves, and grain. Some models (e.g., CERES-wheat) considered winter wheat loss due to
extreme low temperature in winter. In contrast to their strength in representing crop
growth processes, these models have simplified treatment of important upstream
processes that affect crop growth. For example, the photosynthesis scheme is a linear
function of intercepted photosynthetically active radiation (PAR), PAR itself is simplified
as a constant fraction of incoming solar radiation, and radiation is not separated into
direct and diffuse fractions. Further, these crop models were originally developed to
simulate individual, as opposed to multiple crops, making multi-crop simulations at
regional and global scales difficult.
To incorporate more physical processes and to simulate crop growth at regional or global
scales, some agronomic crop growth models were incorporated into agro-ecosystem
models. For example, CERES maize and wheat growth were added into the Decision
Support System for Agrotechnology Transfer Model (DSSAT) (Jones et al., 2003). A
substantial modified version of CERES Wheat (Keating et al., 2001) also has been added
into the Agricultural Production Systems Simulator (APSIM) Model (Keating et al.,
2003). As the effects of vegetation on the atmospheric boundary layer have been
increasingly appreciated, some land surface models started to also incorporate crop
growth models to not only simulate crop yield, but also to simulate crop growth effects
on surface carbon, water, and energy fluxes. For example, the SUCROS crop growth
model was incorporated to JULES (Van den Hoof et al., 2011) and the STIC crop growth
model was incorporated to ORCHIDEE (Wu et al., 2016). In the recent Agricultural
Model Intercomparison and Improvement Project (AgMIP), these agro-ecosystem models
and land surface models were categorized as Global Gridded Crop Models (GGCM).
The Community Land Model (CLM) (Oleson et al., 2013) is one of the GGCM models
included in AgMIP. It is also a state-of-the-art land surface model used in the Community
Earth System Model (Hurrell et al., 2013) that simulates biogeophysical and
biogeochemical processes on a spatial grid. CLM can be run online, coupled with the
atmosphere model, or offline at multiple spatial scales (site, regional, and global) and
resolutions. One grid cell in CLM is divided into different land units (urban, glacier, lake,
wetland, vegetation), and the vegetation unit can consist of up to 14 natural vegetation
types and 64 crop types in the most recent version (a developer version of CLM4.5).
CLM is a community effort that incorporates scientific advances through time, such as
two-leaf stomatal conductance and photosynthesis, transient land use, multilayer canopy
models (Bonan et al., 2012), methane models (Riley et al., 2011), and carbon isotope
models (Koven et al., 2013).
In order to better represent agricultural ecosystems, Levis et al. (2012) introduced crop
growth modules into CLM based on the AgroIBIS model (Kucharik, 2003). Since their
introduction, the crop modules in CLM have been updated to represent more crops types
(maize, soybean, cotton, wheat, rice, sugarcane, tropical maize, tropical soybean) and
processes, such as soybean nitrogen fixation (Drewniak et al., 2013) and ozone impacts
on yields (Lombardozzi et al., 2015). In CLM, crop growth depends on photosynthetic
processes, which are limited by light, water, and nutrient availability. At each time step,
photosynthesis estimations provide the potential available carbon for plant growth, which
is adjusted by nitrogen supply and demand. The actual available carbon is distributed to
leaf, stem, root, and grain by carbon allocation coefficients that vary based on crop
growth stages. While the initial focus for incorporating crop growth into CLM was as a
lower boundary condition to the atmosphere, the model also predicts crop yields and is
participating in the AgMIP GGCM Intercomparison project (Elliott et al., 2015).

Although Levis et al.'s (2012) initial crop growth modules in CLM included a simplified
representation of winter wheat growth, it has never been validated and some of the key
winter wheat growth processes are out of date, such as vernalization, or not included
(e.g., frost tolerance and damage). Our new winter wheat model adopted the same
phenology phases as the original winter wheat model in CLM, but replaced the
vernalization process, added frost tolerance and damage processes, slightly modified the
carbon allocation algorithm, and calibrated several key parameters that affect winter
wheat growth. Our work focused on improving the representation of the key growth
processes for winter wheat in order to, 1) better simulate the land surface influence on
surface $CO_2$, water and energy exchanges in winter wheat-dominated regions, and 2)
accurately simulate crop growth and yield so the model can be used for winter wheat
yield projections.
Methods
Calibration data
We calibrated the simulated leaf area index and yield using observations from the
Atmospheric Radiation Measurement Southern Great Plains Central Facility site (US-
ARM) in northern Oklahoma, USA. The site has well-documented crop growth and
management information, including crop types, planting and harvest dates. The site
conducts bi-weekly leaf area index (LAI) measurements with a light wand (Licor LAI-
2000) during the active growing season. Using a combination of *in situ* LAI and site
reflectance spectrum measurements, Williams and Torn (2015) generated a daily LAI
product, used here to develop and calibrate the winter wheat model. Six winter wheat
seasons are used from the US-ARM site: 2003, 2004, 2006, 2007, 2009, and 2010 (winter
wheat was not grown at the US-ARM site during 2005 and 2008).
Validation data
We validated the simulated leaf area index, and leaf, stem, and grain dry weight at five
winter wheat field sites (TXLU, KSMA, NESA, NDMA, and ABLE) in North America.
The experiments were originally designed to understand winter wheat response to
nitrogen fertilization and water treatments (4 nitrogen levels and 3 irrigation regimes) in
the Great Plains (Hubbard et al., 1988; Major et al., 1988; Reginato et al., 1988), and
have been used as part of the AgMIP Wheat project. For our validations, we only
validated to seven site-year rainfed plots, which are TXLU-1985&1986, KSMA-1985,
NESA-1985&1986, NDMA-1986, and ABLE-1986.
We validated the simulated energy, water, and $CO_2$ flux at three additional eddy flux
tower sites: (1) Ponca City (US-PON), (2) Curtice Walter-Berger Cropland (US-CRT),
and (3) the Washington State University Cook Agronomy Farm conventional tillage site
(CAF-CT) (Figure 1). These three sites do not have detailed crop growth measurements
of tissue biomass, but have surface flux measurements that are crucial to understanding
the role of winter wheat in altering land-atmosphere interactions. One caveat of using the
eddy flux observation is the energy balance closure problem (Foken, 2008; Wilson et al.,
2002) due to the eddy covariance technique limitation or the errors in calculating energy
fluxes terms. The energy closure ratio at the four eddy flux sites are 87% at US-ARM,
91% at US-PON, 70% at US-CRT, and 83% at CAF-CT during the period used in the
comparison. We used these imbalanced energy fluxes as is and discussed their impacts on
our results.
We also validated the simulated US winter wheat yield with the USDA NASS county
level non-irrigated winter wheat yield data. For the sites that did not have site-level yield
observations, we also validated site-level simulations with the nearest county non-
irrigated yield.
Table 1. Winter wheat validation site descriptions.

| Site | Latitude | Longitude | MAT ($^o$C) | Prec (mm) | Simulation years | References |
|---|---|---|---|---|---|---|
| US-ARM | 36.61 | -97.49 | 14.76 | 843 | 2002-2010 | (Fischer et al., 2007; Raz-Yaseef et al., 2015) |
| US-PON | 36.77 | -97.13 | 14.94 | 866 | 1997-1999 | (Hanan et al., 2005; Hanan et al., 2002) |
| US-CRT | 41.63 | -83.35 | 10.10 | 849 | 2012-2013 | (Chu et al., 2014) |
| CAF-CT | 46.78 | -117.08 | 8.74 | 455 | 2013-2014 | (Waldo et al., 2016) |
| TXLU | 33.63 | -101.83 | 8.2 | 531 | 1984-1986 | (Hubbard et al., 1988; Major et al., 1988; Reginato et al., 1988) |
| KSMA | 39.09 | -96.37 | 11.7 | 922 | 1984-1986 | |
| NESA | 41.37 | -100.49 | 11.5 | 499 | 1984-1986 | |
| NDMA | 46.46 | -100.55 | 14.2 | 496 | 1984-1986 | |
| ABLE | 49.42 | -112.5 | 12.2 | 378 | 1984-1986 | |


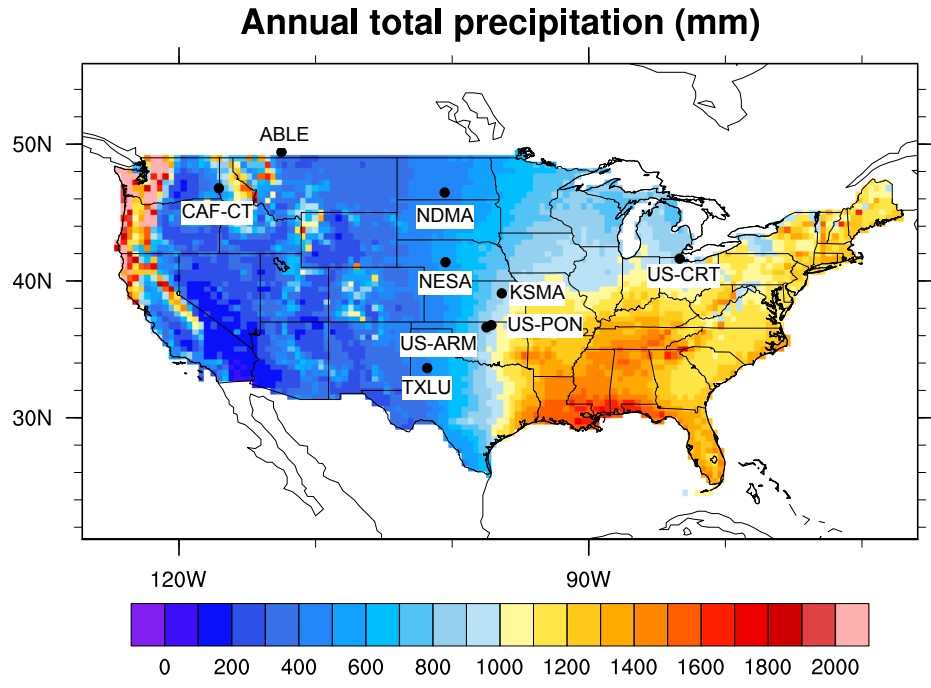

**Annual total precipitation (mm)**

Figure 1. The PRISM 1981-2013 averaged annual total precipitation (mm yr$^{-1}$) and the
nine site locations (US-ARM, US-PON, US-CRT, CAF-CT, ABLE, NDMA, NESA,
KSMA, TXLU) used in this study.
*Model development*
Similar to other crops in CLM, winter wheat has four phenological phases, including
planting, leaf emergence, grain fill, and harvest. The criteria and thresholds for entering
different phenology phases are listed in Table 2. Growing degree days is the key variable
controlling phenology, and is measured as heat accumulation during the whole growing
season or over a certain period. It was calculated by accumulating the difference (no
accumulation if less than 0) between the target temperature (e.g., mean air temperature)
and base temperature, and normally has a maximum daily increment. We used three
different growing degree day algorithms to determine winter wheat phenology, all using
the same base temperature (0 $^{o}$C) and maximum daily increment (26$^{o}$) (Levis et al.,
2012). The 20-year running average of growing degree days (GDD$_{020}$) uses 2-meter air
temperature (T$_{2m}$) from September to June in the northern hemisphere (from April to
September in Southern Hemisphere), and is updated each year by averaging the previous
19 years. The growing degree days for soil temperature since planting (GDD$_{tsoi}$) uses
averaged soil temperature from the top two model soil layers (0.71 cm and 2.79 cm).
Growing degree days since planting (GDD$_{plant}$) uses T$_{2m}$, and is reduced by a
vernalization factor (see below) after leaf emergence.
Table 2. Criteria and notation for winter wheat to enter each phenological stage.

|  | Criteria | Notation |
|---|---|---|
| Planting | 5 day running minimum temperature < minimum | $T_{5d} < 5\,{}^{\circ}C$ |

| | planting temperature and, day of year > minimum planting day of year and, 20-year running average of gdd0 > minimum gdd | $doy > 1^{st}Sep$ $GDD_{020} > 50$ |
|---|---|---|
| Leaf emergence | Growing degree days of soil temperature to 2.79cm depth > 3% of maturity growing degree days | $GDD_{tsoi}$ $> 3\%GDD_{mat}$ |
| Grain fill | Growing degree days of 2m temperature since planting > 40% of maturity growing degree days | $GDD_{plant}$ $> 40\%GDD_{mat}$ |
| Harvest | Growing degree days of 2m temperature since planting ≥ maturity growing degree days or, the number of days past planting > maximum growing days | $GDD_{plant} \geq GDD_{mat}$ $DPP > 330$ |


To better represent winter wheat phenology, we added two additional processes:
vernalization and frost damage. We adopted a generalized winter wheat vernalization
model (equation 1-3 were directly adopted from Streck et al., (2003)). Similar to other
winter crops, winter wheat must be exposed to low and nonfreezing temperature to enter
the reproductive stage. Additionally, the vernalization process affects cold tolerance, as
discussed below. If plants are not fully vernalized, the potential size of the flower head
will be reduced. Vernalization starts after leaf emergence and ends before flowering. To
model this process, daily vernalization rate (fvn, eq. 1) is calculated based on the
difference between the crown temperature ($T_{crown}$) and the optimum vernalization
temperature ($T_{opt}$). In the CLM crop model, the crown temperature is the crown depth soil
temperature (Aase and Siddoway, 1979), calculated as the function of 2-meter air
temperature and snow depth. The crown temperature is typically warmer than the 2-meter
air temperature in winter, if the plant is covered by snow, and the same as the 2-meter air
temperature without snow cover. If the crown temperature is equal to the optimum
temperature for a whole day, then fvn is equal to 1. Otherwise, fvn is less than 1 as
calculated in eq. 1.


$fvn(T_{crown}) =$
$$\begin{cases} \dfrac{\left[2(T_{crown}-T_{min})^{\alpha}(T_{opt}-T_{min})^{\alpha}-(T_{crown}-T_{min})^{2\alpha}\right]}{(T_{opt}-T_{min})^{2\alpha}} & T_{min} \leq T_{crown} \leq T_{max} \\ 0 & T < T_{min} \text{ or } T_{crown} > T_{max} \\ 1 & T_{crown} = T_{opt} \end{cases} \quad \text{(eq. 1)}$$


$where \ \alpha = \dfrac{ln2}{\ln[(T_{max} - T_{min})/(T_{opt} - T_{min})]}$


Next, the sum of *fvn* over sequential days is the effective vernalization days (*VD*, eq. 2).

$VD = \sum fvn(T_{crown})$    (eq. 2)

This is used to calculate the vernalization factor (*VF*, eq. 3). VF varies from 0 to 1 (fully
vernalized) to represent the vernalization stage.

$VF = \frac{VD^5}{22.5^5+VD^5}$ (eq. 3)

Finally, VF was used in adjusting the growing degree days since planting
($GDD_{plant}=GDD_{plant,unadjusted} \times VF$) and the grain carbon allocation coefficient ($a_{grain} =$
$a_{grain,unadjusted} \times VF$). When winter wheat is not fully vernalized (*VF* < 1) then $GDD_{plant}$
and $a_{grain}$ are reduced, resulting in slowed growth and reduced yield.

We quantify the impacts of low temperature damage, including from frost, using three
variables: 1) temperature at which 50% of winter wheat was damaged ($LT_{50}$), 2) survival
probability (fsurv), and 3) winter killing degree days (WDD). Here, equation 4-8 were
from Bergjord et al., (2008) and equation 9-10 were from Vico et al., (2014), without any
modifications. The calculations for the three variables are briefly summarized, and more
detailed descriptions of the calculations can be found in Bergjord et al., (2008) and Vico
et al., (2014). $LT_{50}$ (eq. 4) depends on $LT_{50}$ from the previous time step ($LT_{50t-1}$), low
temperature acclimation (i.e. hardening; RATEH), loss of hardening due to exposure to
high temperatures (i.e. dehardening; RATED), stress due to respiration under snow
(RATER), and exposure to low temperature (RATES). Lower $LT_{50}$ results in greater frost
tolerance for winter wheat while higher $LT_{50}$ indicates lower frost tolerance.


$LT_{50t} = LT_{50t-1} - RATEH + RATED + RATES + RATER$ (eq. 4)

$RATEH = H_{param}(10 - \max(T_{crown}, 0))(LT_{50t-1} - LT_{50c})$ $T_{crown} < 10°C$ (eq. 5)


The contribution of hardening to $LT_{50}$ was calculated as RATEH (eq. 5), which was
mainly a function of crown temperature ($T_{crown}$) and adjusted by a hardening parameter
($H_{param}=0.0093$), maximum frost tolerance ($LT_{50c}=-23$ °C). RATEH increased rapidly
when crown temperature ($T_{crown}$) fell below 10 °C. When $T_{crown}$ fell below 0 °C, the slope
of RATEH was same as $T_{crown}$ at 0 °C. RATEH is also determined by the difference
between the current level of frost tolerance and the maximum level of frost tolerance
($LT_{50t-1} - LT_{50c}$). At the beginning of cold acclimation, when $LT_{50t-1}$ is much higher
than $LT_{50c}$, RAHEH increases quickly.

$RATED = D_{param}(LT_{50i} - LT_{50t-1})(T_{crown} + 4)^3$ $\begin{array}{l} T_{crown} \geq 10°C\ when\ VF < 1 \\ T_{crown} \geq -4°C\ when\ VF = 1 \end{array}$ (eq.
292 6)
$where\ LT_{50i} = -0.6 + 0.142LT_{50c}$ represents LT50 for an unacclimated plant

RATED accounts for the dehardening contribution (eq. 6), which is a function of crown
temperature and is adjusted by a dehardening parameter ($D_{param}=2.7 \times 10^{-5}$) and $LT_{50}$ for a
plant that is not acclimated to cold ($LT_{50i}$). Cold acclimation is a cumulative process and
can reverse (dehardening) when plants are exposed to high temperature or restart
(hardening) when temperature is below 10 $^{\circ}$C. The high temperature threshold depends
on the vernalization stage. Dehardening occurs when $T_{crown} \geq 10°C$ for plants that are
not fully vernalized (VF<1), and when $T_{crown} \geq -4°C$ for plants that are fully vernalized
(VF=1).
$RATER = R_{param} \times RE \times f(snowdepth)$ (eq. 7)
$where\ RE = \dfrac{e^{0.84+0.051T_{crown}-2}}{1.85}, R_{param} = 0.54$
$f(snowdepth) = \min(snowdepth, 12.5)/12.5$
Stress due to respiration under snow also increases $LT_{50}$ and was calculated as RATER
(eq. 7), which is a function of snow depth and a respiration factor (RE). RE is a
regression function fitted to respiration measurements (Sunde, 1996). $f(snowdepth)$
ranges from 0 to 1 for snow depth up to 12.5cm, and is equal to 1 when snow depth is
greater than 12.5cm.
$RATES = \dfrac{LT_{50t-1}-T_{crown}}{e^{-S_{param}(LT_{50t-1}-T_{crown})-3.74}}$ (eq. 8)
$where\ S_{param} = 1.9$
Long-term exposure to near lethal temperature will also increase $LT_{50}$ and was calculated
as RATES (eq. 8), which is based on the winter survival model developed by (Fowler et
al., 1999).
The probability of survival (fsurv, eq. 9) is a function of $LT_{50}$ and crown temperature.
The probability of survival reaches a median value when $T_{crown}$ equals $LT_{50}$, and
increases when $T_{crown}$ is warmer than LT50 and decreases when $T_{crown}$ colder than $LT_{50}$.
$f_{surv}(T_{crown}, t) = 2^{-(\frac{|T_{crown}(t)|}{|LT50(t)|})^{\alpha surv}}$    $T_{crown} \leq 0°C$   (eq.9)
Finally, we calculate winter killing degree days (WDD, eq. 10) as a function of $T_{crown}$ and
*fsurv*. WDD not only accounts for the cumulative degree days when the crop was
exposed to freezing temperatures but also accounts for the probability of death at the
temperature of exposure. High WDD occurs with low temperature and low survival
probability.
$WDD = \int_{winter} \max[(T_{base} - T_{crown}), 0] [1 - f_{surv}(T_{crown}, t)]\ dt$  (eq. 10)
$where\ T_{base} = 0°C$
Although Bergjord et al. (2008) and Vico et al. (2014) defined the frost tolerance and
damage indicators described above, they did not propose a model for the growth response
to crop damage from low temperatures. Here we developed our own hypothetical two-
stage frost damage parameterization (equation 11-12) that includes both instant damage
and accumulated damage during the leaf emergence phase of winter wheat growth. In
CLM, plants tissues are represented as the mass of carbon and nitrogen per $m^2$ ground.
We simulated leaf carbon and nitrogen reduction for each of the two types of frost
damage. We assumed that instant damage occurs at the beginning of the growing season
($VF$<0.9) when plants are not fully vernalized and have low survival probability when
exposed to subzero temperatures. In this case, the growth of leaves most vulnerable to
cold (e.g., new leaves or small seedlings) would slow or cease. After many sensitivity
tests, we found the best fit to observations by removing an amount of leaf carbon
($leafc_{damage\_i} = 5$ g C/$m^2$) to the soil carbon litter pool, scaled by a factor of 1-*fsurv* (eq. 11)
at each time step (half-hourly). The leaf carbon was reduced whenever *fsurv* was less
than 1 until leaf carbon reached a minimum value (10 g C/$m^2$).

$leafc_t = leafc_{t-1} - leafc_{damage\_i}(1 - fsurv)\,, for\ WDD > 0, fsurv < 1,$
$and\ leafc_t > 10$ (eq. 11)

In addition to this instantaneous damage, we introduced an accumulated damage
parameterization for when winter wheat is close to or has completed vernalization
($VF$>0.9) in spring. We assumed that plants would not be likely to suffer as much from
instantaneous frost damage as in the early winter season due to less subzero temperature,
but that an extended period of subzero temperatures (large WDD) would lead to severe
crop damage. To simulate this, we let WDD accumulate up to a set value (set to $1^o$ days),
when it triggers the accumulated damage function and we track the average *fsurv* for this
time period. When WDD>$1^o$ days, all leaf carbon from previous time step ($leafc_{t-1}$,
representing the damage to the whole plant), scaled by a factor of (1- *averaged fsurv*),
was removed from the leaf carbon to the soil carbon litter pool. After leaf carbon was
reduced, *WDD* was reset to 0, and the accumulation and tracking of the averaged *fsurv*
was restarted. For both frost damage types, leaf nitrogen was removed to the nitrogen
litter pool. The nitrogen was scaled to the reduction of leaf carbon by the fixed C:N ratio
(25 for winter wheat). The results show that the simulation of LAI (Figure S1) can be
improved by including a representation of frost damage in winter wheat models.
However, the approach here is based on empirical indicators of frost damage. This
suggests the potential for further improvement by incorporating process-level
representation of frost damage in future model versions.

$leafc_t = leafc_{t-1} \times averaged\ fsurv,\ \ VF \geq 0.9\ and\ WDD > 1$    (eq. 12)

CLM leaf ($a_{leaf}$) and stem ($a_{livestem}$) carbon allocation coefficients for winter wheat were
also adjusted during the grain fill to harvest phase. The original $a_{leaf}$ and $a_{livestem}$ changed
in time as a function of growing degree days. This approach resulted in a rapid decline in
the stem carbon allocation, and led to a grain carbon allocation coefficient that was too
large (Figure S2), producing unrealistically high yields at the US-ARM site. We modified
the leaf and stem carbon allocation coefficients to be functions of carbon allocation at the
initial time of grain fill ($a_{leaf}^{i,3}$ and $a_{livestem}^{i,3}$), and therefore $a_{livestem}$ gradually declines and
$a_{grain}$ gradually increases during the grain fill phase (Table 3, Figure S2b).
After the above modification of carbon allocation and addition of the new vernalization
and frost damage processes, we calibrated three parameter values (indicated with * in
Table 4) in the US-ARM simulation. We adjusted minimum planting temperature and
maximum days for growing to better match the US-ARM site planting and harvest date,
and adjusted the initial leaf carbon allocation coefficient to better match the US-ARM
LAI and yield.
Table 3. Carbon allocation algorithms for the leaf emergence to grain fill stage, and the
grain fill to harvest stage.

| Phase | Allocation algorithm |
|---|---|
| Leaf emergence to grain fill | $a_{grain} = 0$ <br> $a_{froot} = a_{froot}^i - (a_{froot}^i - a_{froot}^f)\dfrac{GDD_{T_{2m}}}{GDD_{mat}}$ <br> $a_{leaf} = (1 - a_{froot})\dfrac{f_{leaf}^i(e^{-0.1} - e^{[-0.1(GDD_{T_{2m}}/h)]})}{e^{-0.1} - 1}$ <br> $a_{livestem} = 1 - a_{grain} - a_{froot} - a_{leaf}$ |
| Grain fill to harvest | $a_{leaf} = a_{leaf}^{i,3}$ when $a_{leaf}^{i,3} \le a_{leaf}^f$ else <br> $a_{leaf} = a_{leaf}^{i,3}(1 - \dfrac{GDD_{T_{2m}} - h}{GDD_{mat}d_L - h})^{d_{alloc}^{leaf}}$ <br> $a_{livestem} = a_{livestem}^{i,3}$ when $a_{livestem}^{i,3} \le a_{livestem}^f$ else <br> $a_{livestem} = a_{livestem}^{i,3}(1 - \dfrac{GDD_{T_{2m}} - h}{GDD_{mat}d_L - h})^{d_{alloc}^{stem}}$ <br> $a_{froot} = a_{froot}^i - (a_{froot}^i - a_{froot}^f)\dfrac{GDD_{T_{2m}}}{GDD_{mat}}$ <br> $a_{grain} = 1 - a_{livestem} - a_{froot} - a_{leaf}$ |

Table 4. A list of key parameters used for phenology and carbon and nitrogen allocation
for the original and modified CLM winter wheat models.

| | Parameters | Description | Original | Modified |
|---|---|---|---|---|
| Phenology | *minplanttemp | Minimum planting temperature | 278.15 (K) | 283.15 (K) |
| | *mxmat | Maximum days for growing | 265 (days) | 330 (days) |
| | $GDD_{mat}$ | Maturity growing degree days | 1700 | 1700 |
| | gddmin | Minimum growing degree days for planting | 50 | 50 |
| | lfemerg | Percentage of gddmaturity to enter leaf emerge phase | 3% | 3% |
| | grnfill | Percentage of gddmaturity to enter grain fill phase | 40% | 40% |
| CN | $a_{froot}^i$ | Initial value of root carbon allocation coefficient | 0.3 | 0.3 |
| | $a_{froot}^f$ | Final value of root carbon allocation coefficient | 0 | 0 |
| | * $f_{leaf}^i$ | Initial value of leaf carbon allocation coefficient | 0.425 | 0.6 |

| | | | | |
|---|---|---|---|---|
| $h$ | Heat unit threshold (grnfill x hybgdd) | 680 | 680 |
| $d_L$ | Leaf are index decline factor | 1.05 | 1.05 |
| $d_{alloc}^{leaf}$ | Leaf carbon allocation decline factor | 3 | 3 |
| $d_{alloc}^{stem}$ | Stem carbon allocation decline factor | 1 | 1 |

[*]indicates parameters that have different values between original and modified model.
*Experiment design*
We set up paired CLM4.5 site simulations using Levis et al.'s (2012) original winter
wheat model (CLMBASE) and our modified winter wheat model (CLMWHE) at the
winter wheat sites in Table 1. We forced the site simulations with half-hourly observed
temperature, relative humidity, precipitation, wind, and incoming solar radiation.
Incoming longwave radiation was available at the US-ARM and US-CRT sites and was
also input to the simulations at those sites. Each paired simulation ran with the same
initial conditions, which were generated using a spin-up of several hundred years at each
site (described below). The simulated differences between the original winter wheat and
the modified winter wheat are therefore due to the modified parameters and updated
processes described above.
Land surface models, especially those including biogeochemical components, require
long-term (thousands of simulation years) spin-up for their carbon and nitrogen pools to
reach equilibrium (Shi et al., 2013). Therefore, generating initial conditions with steady-
state carbon and nitrogen pools is computationally time consuming and expensive if the
simulation starts with no carbon and nitrogen. To accelerate the spin-up process, we
generated site-level initial conditions by interpolating a global simulation that had
reached carbon and nitrogen equilibrium, and then further spun up the site-level
simulations for 200 years using recycled site observed meteorology for years listed in
Table 1. When CLM reaches equilibrium, the averaged land surface variables during each
atmospheric forcing cycle should not change or vary within a threshold (Table S1). We
found latent heat flux, sensible heat flux, leaf area index, and wheat yield reached
equilibrium fairly quickly (<40 years), but the total ecosystem carbon, total soil organic
carbon, and total vegetation carbon took a longer time to reach the equilibrium state.
We also set up a regional simulation (50km resolution, 1979-2010) over the continental
U.S. to compare spatial patterns in yield predictions to the USDA NASS county level
winter wheat yield. To get the winter wheat land cover percentage, we first estimated the
winter wheat fraction using the USDA NASS county level acres harvested data, and then
split the wheat land cover percentage in the default CLM surface file into winter wheat
and spring wheat. Since the goal of the regional simulation was to validate the spatial
yield and not the carbon pools, we ran a partial spin-up and allowed the crop yield to
reach equilibrium while the total ecosystem (i.e., soil) carbon was not at equilibrium.
We applied the nitrogen fertilization in all the simulations. CLM4.5 considered the
nitrogen limitation through the down regulation of the potential photosynthesis based on
the nitrogen demand and supply deficit, which was calculated by considering the
complex below ground biogeochemical processes (e.g., nitrification, denitrification,
leaching, soil organic matter decomposition). When nitrogen supply is less than the
nitrogen demand, the potential photosynthesis will be reduced by the deficit factor. For
the TXLU, KSMA, NESA, NDMA, and ABLE site simulations, we applied the observed
nitrogen fertilization amount (10-20 gN/m$^2$) at the same days as the observation. While
for the other sites and the US simulations, we applied the default nitrogen fertilization
during leaf emergence every year for an amount of 8gN/m$^2$. With these nitrogen
fertilization, there are no nitrogen limitation at all our simulations.
*Statistical analysis of yield at US-ARM site*
To determine the factors that contributed most strongly to yield in observations and the
model, we performed statistical regressions for US-ARM observations and CLMWHE
outputs separately. We had 11 observed and simulated variables including growing
degree days, nitrogen fertilization, peak leaf area index, precipitation, days of grain fill,
days of leaf emergence, day of peak leaf area index, 10cm soil moisture, 20cm soil
moisture, planting date, and harvest date. We performed simple linear regressions with
each of these variables and compared the R2 values between observational data and
simulation outputs.
Results
*Leaf area index and dry weight*
The modified winter wheat model (CLMWHE) showed a better seasonal growth cycle
than the original model (CLMBASE) (Figure 2). In the CLMBASE simulation, the
vernalization factor is too high even at the beginning of the growing season (Figure S3).
Without the reduction on the growing degree days from the vernalization function, winter
wheat LAI and leaf weight reached peaks in December and then declined due to the onset
of the grain fill stage. The long grain fill stage (December – May) in CLMBASE did not
produce a sufficiently high grain mass because of the low rate of photosynthesis with the
low LAI. In the CLMWHE simulation, LAI and leaf weight reached peaks in April, and
stem and grain weight reached peaks in May, which are similar to the site observations.
The improvements in the seasonal variation are mainly due to the updated vernalization,
which produced a reasonable vernalization period about two-three months, reduced the
growing degree days and extended the leaf emergence stage. The cold damage scheme
also played a role in reasonable simulation of winter LAI and leaf weight. For example,
at KSMA-1985, cold damage reduced the LAI and leaf weight in fall yielding a better
match to the winter measurement (at DOY=320). Besides these improvements, we also
observed an overestimation of LAI during the later growing season, which is due to the
low leaf senescence rate during the grain fill period.
The updated winter wheat model captured the grain weight temporal and spatial
variations, and RMSE and the index of agreement are better in CLMWHE than
CLMBASE for seven site-years. RMSE was reduced by 19% and index of agreement was
increased by 45%. CLMWHE showed higher grain weight in 1986 than 1985 at TXLU
and NESA, as did the observations, because 1986 was a wetter year for both TXLU (8%
higher annual precipitation than 1985) and NESA (84% higher). In 1986, CLMWHE
showed more grain weight in NESA and NDMA than TXLU and ABLE, as in the
observations.

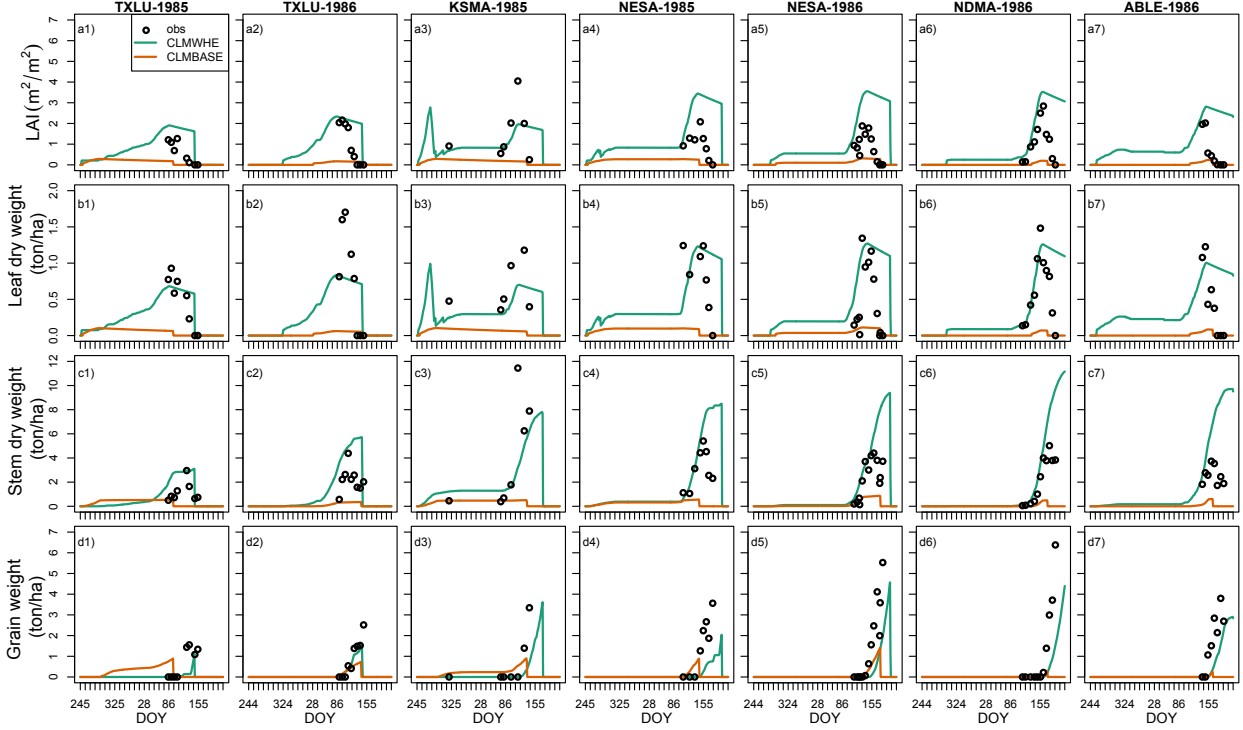

Figure 2. The daily leaf area index ($m^2/m^2$), leaf dry weight (ton/ha), stem dry weight
(ton/ha), and grain dry weight (ton/ha) simulations in CLMWHE (the updated winter
wheat model) and CLMBASE (the original winter wheat model), and in site observations
for seven site-years.
For the four flux tower sites, CLMWHE also improved LAI and crop growth seasonal
variations (Figure 3a-d). Both sites exhibited reduced RMSE compared to CLMBASE
(Table S3). At the US-ARM site, CLMWHE underestimated peak LAI but captured the
seasonal LAI variation (peak in April and then decline). At the US-PON site, CLMWHE
overestimated LAI throughout the growing season but showed similar seasonal variation.
Although US-CRT and CAF-CT sites have no LAI observations, CLMWHE generally
increased LAI and had a more reasonable seasonal variation compared to CLMBASE.
*Surface carbon, water and energy fluxes*
The improved simulation of LAI seasonal variation led to better monthly patterns of net
ecosystem exchange of $CO_2$ (NEE) (Figure 3e-h). In Figure 3, negative values indicate a
carbon sink, where the crop gains more carbon through photosynthesis than is lost due to
respiration. During the winter wheat growing season, the observed NEE is most negative
coincident with peak LAI. CLMWHE captured these seasonal patterns at US-ARM and
US-CRT sites, although it did underestimate the NEE magnitudes at their peak. The
underestimation of peak LAI may have contributed to this bias. CLMBASE has much
smaller NEE relative to CLMWHE, consistent with the lower LAI. We also observed a
discrepancy after harvest, where CLMWHE (and CLMBASE, to a lesser extent)
simulated a strong carbon source for the site, but observations exhibited either neutral
NEE at US-ARM or a smaller NEE at US-CRT site. This discrepancy is due to the model
treating the land cover as bare ground after harvest, when in reality weeds (identified by
visual inspection of daily site photographs) quickly exert influence on surface fluxes of
carbon.
The annual net radiation (Rn) simulations (Figure 3i-l) at the four sites were slightly
improved in CLMWHE. Averaged across the four sites, Rn RMSE was reduced from
16.6 $W.m^{-2}$ in CLMBASE to 12.9 $W.m^{-2}$ in CLMWHE. The latent heat flux (LE)
simulation was improved during March-May (Figure 3m-p). The spring LE RMSE was
reduced by 10-70% across the four sites in CLMWHE due to the better LAI simulation in
spring. However, the annual LE RMSE was only slightly reduced (up to 23% RMSE
reduction in CLMWHE) at US-ARM, US-PON, and US-CRT, and showed no
improvement at CAF-CT. The sensible heat flux (H) showed no obvious improvement
(Figure 3q-t).

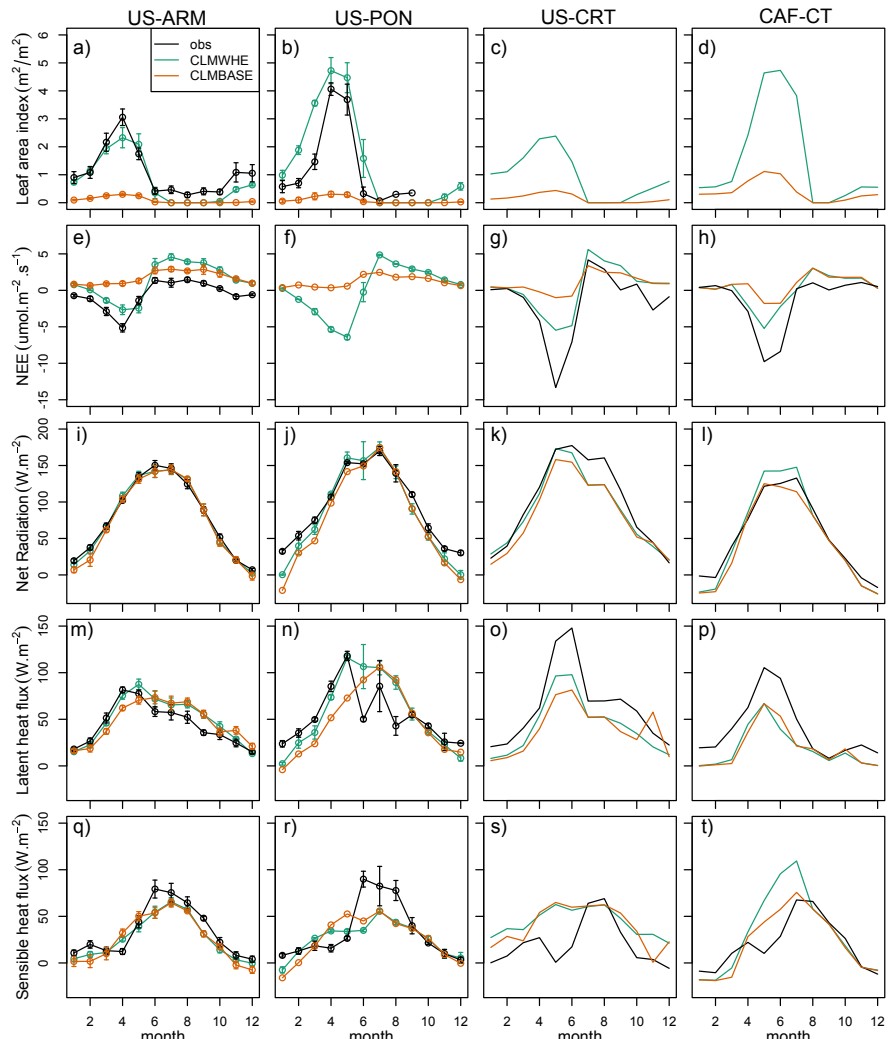

Figure 3. Monthly averaged (a)-(d) leaf area index ($m^2/m^2$), (e)-(h) net ecosystem
exchange of $CO_2$ ($umol.m^{-2}.s^{-1}$), (i)-(l) net radiation ($W.m^{-2}$), (m)-(p) latent heat flux
($W.m^{-2}$), and (q)-(t) sensible heat flux ($W.m^{-2}$) for observations, CLMWHE, and
CLMBASE across four sites. The US-ARM site data were averaged over six winter
wheat years (2003, 2004, 2006, 2007, 2009, 2010), US-PON data was averaged over
1997 and 1998, US-CRT data is from 2013, and CAF-CT data is from 2014. The error
bars indicate the standard error for the month across years, and there are no error bars for
US-CRT and CAF-CT because the values are for one year.
At the US-ARM and US-PON sites, the LE monthly variation patterns were improved by
better representing leaf area index, but this improvement was limited by surface energy
partitioning problems in the model. The model partitioned more energy to LE than was
observed during the period when LAI declines in the late growing season (May-July).
The observed LE is 45% and 53% of net radiation at US-ARM and US-PON site, while
LE simulated in CLMWHE is 53% and 67% of net radiation at US-ARM and US-PON
site. This energy partitioning problem is reversed at the US-CRT and CAF-CT sites,
where the model partitioned less energy to LE than observations. The observed LE is 68%
and 66% of net radiation at US-CRT and CAF-CT sites, while simulated LE in
CLMWHE is 52% and 30% of net radiation at US-CRT and CAF-CT site. Both sites are
rainfed with no irrigation applied. In addition, the month of peak LE does not coincide
with the month of peak LAI in the observations at US-ARM and US-PON. In
observations, LE reaches a peak at the same time when LAI is at its peak, but in
CLMWHE, LE reaches peak one month later than the LAI peak. The lack of energy
balance closure for the eddy flux measurements could affect the energy fluxes RMSE
estimations but will not change the major conclusions here: CLMWHE showed improved
spring LE simulations than CLMBASE, and the simulated LE peak was one month later
than LAI peaks. Finally, we note that the winter wheat model did not improve surface
energy partitioning in summer after winter wheat harvest.
We found that the overestimation of LE in summer and fall can be reduced using a new
soil evaporation scheme (Swenson and Lawrence, 2014) that will be available in CLM5.
In CLM, vegetation affects LE through leaf transpiration, and LE in vegetated grid cells
has three components: soil evaporation, wet leaf evaporation, and dry leaf transpiration
(Lawrence et al., 2007). The excessive spring soil evaporation in CLM has been reported
in earlier versions of CLM (Lu and Kueppers, 2012; Stockli et al., 2008) and some effort
has been made to reduce soil evaporation. For example, Sakaguchi and Zeng (2009)
added a litter resistance to soil evaporation in CLM3.5 that reduced the annual averaged
soil evaporation. Recent work by Swenson and Lawrence (2014) added a dry surface
layer that increased the soil resistance and reduced soil evaporation. We tested the new
dry surface layer scheme at the US-ARM site, and found that soil evaporation was
reduced by 21% and the LE simulation was improved in May-December (Figure 4c).
However, the spring LE was still underestimated and the LE peak was still one month
later than LAI peak, which is due to the leaf transpiration reaching its peak one month
later than the LAI peak (Figure 4c).

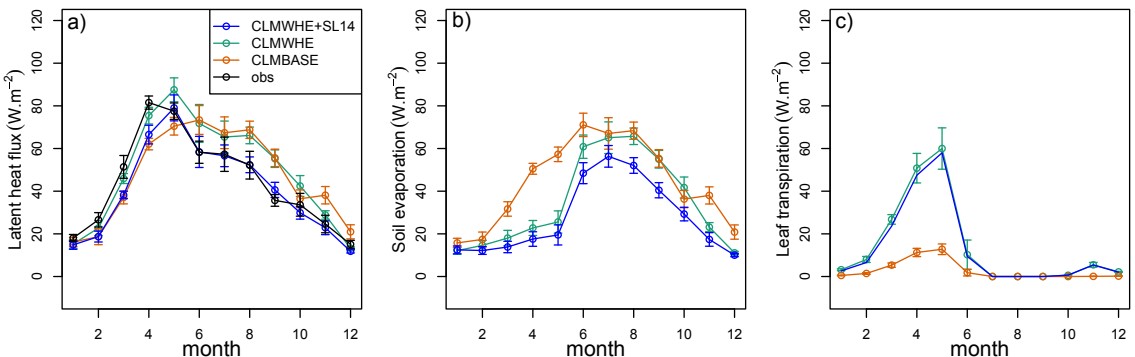

Figure 4. US-ARM site monthly averaged (across six years) a) latent heat flux (W.m$^{-2}$), b)
leaf transpiration (W.m$^{-2}$), and c) soil evaporation (W.m$^{-2}$). CLMWHE+SL14 is the same
simulation as CLMWHE but with the new soil evaporation scheme by Swenson and
Lawrence (2014).
*Yield*

The accuracy of the simulated yield depended on whether the region has a similar climate
as the site where the model was calibrated (Figure 5). US-ARM had the smallest RMSE
(0.80 ton/ha) due to calibration, and US-PON site had only a slightly higher RMSE (1.11
ton/ha) than US-ARM because the two sites have similar climate (both are located in
northern Oklahoma). The yield was overestimated by 0.59 and 1.00 ton/ha for US-ARM
and US-PON. However, at US-CRT and CAF-CT, which are far from US-ARM, the
yield RMSE values were much higher (2.46 and 3.68 ton/ha) and yields were
underestimated by 2.45 and 3.68 ton/ha. In terms of the interannual variation in yield,
CLMWHE accurately simulated the yield decline at the US-ARM site from 2003-2006
and captured the interannual variation from 2007-2010, but failed to simulate the lowest
yield in 2007. We also note that CAF-CT is the only site where yield simulations with
CLMWHE were worse than CLMBASE. Here the yield RMSE increased from 0.90
ton/ha in CLMBASE to 3.86 ton/ha in CLMWHE (discussed further below).

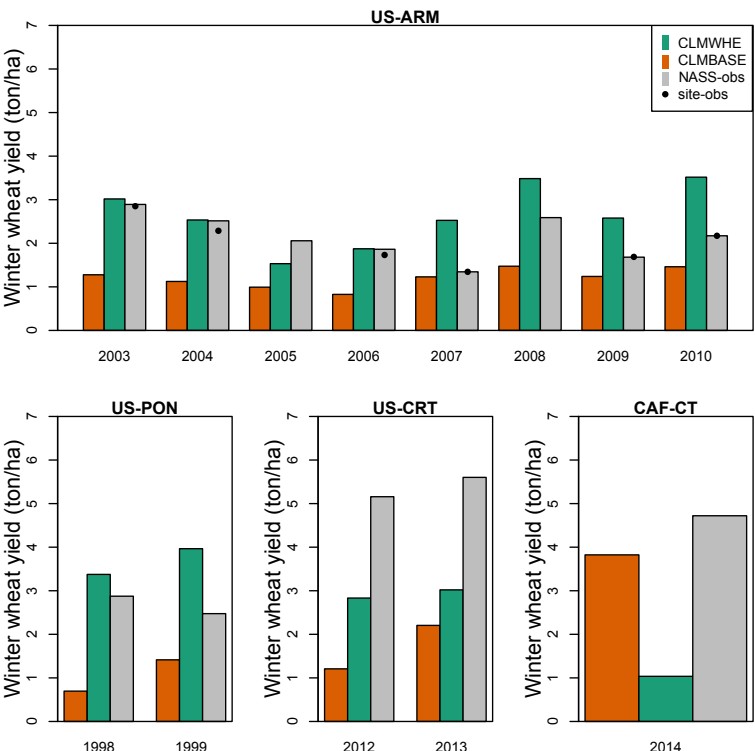

Figure 5. The annual winter wheat yield compared against the nearest county USDA
NASS yield data and site observations (if available). The nearest county USDA NASS
yield data is very similar to the site measured yield at the US-ARM site.

CLMWHE (Figure 6b) showed a better US yield estimation (RMSE reduced by 24%)
than CLMBASE (Figure 6c) but still underestimated the US winter wheat yield by 35%
compared to USDA county level non-irrigated winter wheat yield data averaged across
1979-2010 (Figure 6a), which is largely due to the underestimation of the Northwest US
winter wheat yield. In the simulation, winter wheat growth in the Northwest was limited
by soil water availability. Figure 7 shows that the plant wetness factor (btran, averaged
across growing season) was <0.5 in much of the region. In CLM, btran varies between 0
and 1 and represents the available soil water to the plant (1 means no water stress at all).
The low btran in this region limited photosynthesis and reduced crop yield in the model.
We applied irrigation to a single point in the Northwest, and the yield increased from
1.98 ton/ha to 5.42 ton/ha with irrigation, which is consistent with yields in subregions of
the Northwest. For the Southeast US, CLMWHE simulated a similar yield as the
Southern Great Plains, but the simulated yield was lower than USDA yield for the region,
which may be due to model deficiencies in the representation of fertilization, lack of
regional varieties, or other forms of crop management not well captured in the model.

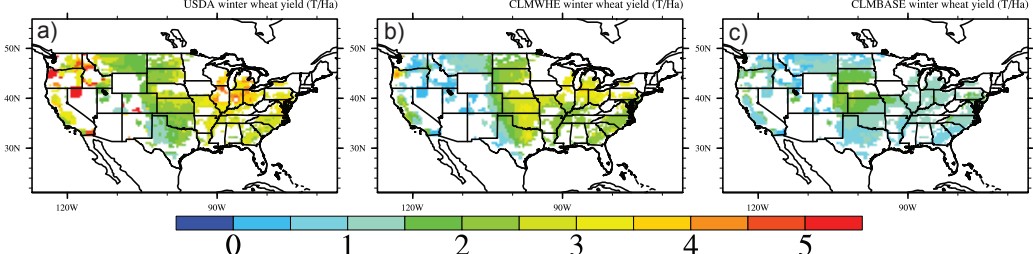

Figure 6. 1979-2010 averaged winter wheat yield for (a) USDA county level yield, (b)
the CLMWHE simulated yield, and (c) CLMBASE simulated yield.

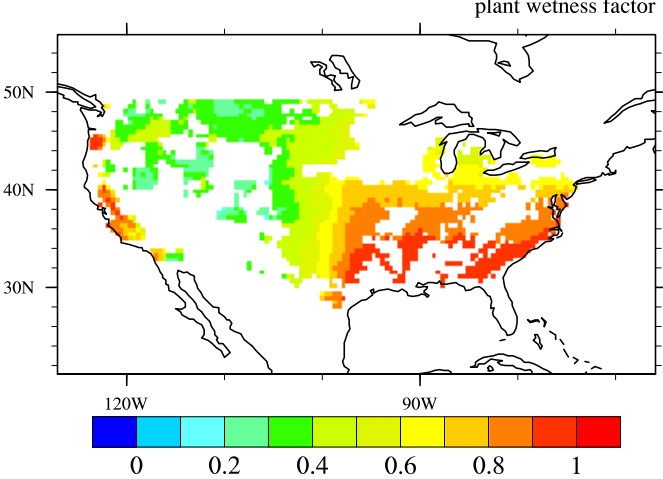

Figure 7. 1979-2010 averaged plant wetness factor between leaf emergence and harvest.
Values less than 1 indicate water stress and cause photosynthesis to be reduced in the
model.
We quantified frost damage impacts on LAI and yield in the US domain through
CLMWHE simulations with and without the frost damage function. Frost damage
resulted in lower LAI and yield, with spatial variation across the U.S (Figure 8). For the
domain average, frost damage reduced LAI by 27% (or 1.69 m$^2$/m$^2$) and reduced yield by
28% (or 0.5 ton/ha). The greatest reduction (>45%) in LAI occurred in Texas and the
southeastern US, which was due to insufficient hardening, producing a high LT50 and
low survival rate. LAI in the cold northern US regions had less impact (<15%) from frost
damage. The cold damage indirectly affects yield through reduced photosynthesis with
lower LAI, but photosynthesis and yield changes were not always geographically
consistent with the LAI damage. For example, the northern Great Plains and Midwest had
greater percentage reductions (>45%) in yield than reductions in LAI (< 15%).

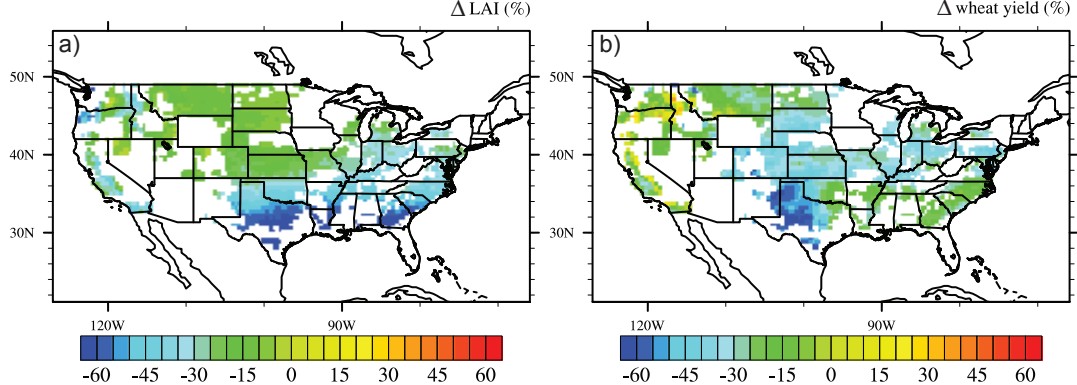

Figure 8. Frost damage-induced percentage difference in (a) leaf area index and (b) yield
between two 1979-2010 CLMWHE simulations, one with frost damage and one without
656                                        frost damage.
A simple, single variable, statistical yield regression indicated that variables important in
predicting CLMWHE yield may be irrelevant for predicting observed yield. The
simulated yields depend most on the growing degree days (R$^2$=0.94), which only
explained 24% of observed yield variation (Figure 9). Although there are many other
variables that contribute to variation in the CLMWHE yield, such as peak LAI, length of
leaf emergence period, harvest date, and day of LAI peak, these variables have strong
correlations with growing degree days, which suggests that crop yields in CLM depend
too much on growing degree days. Soil moisture, especially the lower layer soil moisture
at 20cm, is the only variable that explained a large amount of yield variation in both
observations (R$^2$=0.80) and CLMWHE (R$^2$=0.86). So improved representation of soil
hydrology, especially the interannual variability of soil moisture may improve the
simulations of yield variation.

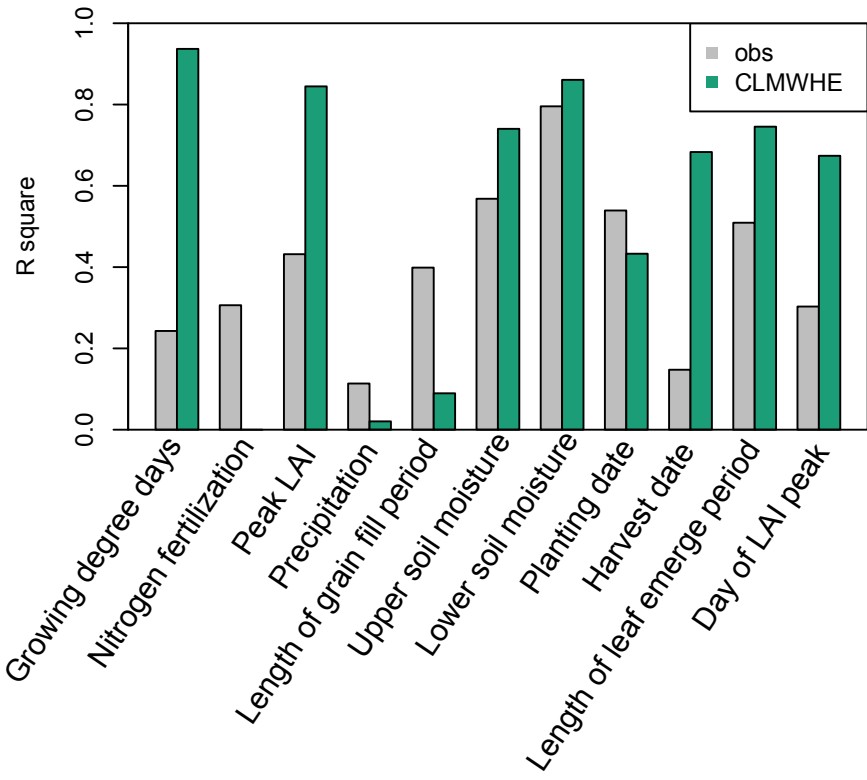

Figure 9. Comparison of the linear regression R square for yield and each of the 11
variables.
Discussion and conclusions
We improved the winter wheat model in CLM with new vernalization, frost tolerance,
and frost damage processes. We modified the grain carbon allocation algorithm and
performed a calibration on three key parameters (minimum planting temperature,
maximum crop growth days, and initial value of leaf carbon allocation coefficient) at the
US-ARM site, and then validated the model performance at multiple other sites in North
America. These model alterations led to large improvements for crop phenology
(indicated by LAI), net ecosystem exchange, and spring latent heat flux. Additionally, the
modeled yield RMSE is comparable to literature values (Palosuo et al., 2011). However,
there are several remaining limitations of the model that need to be resolved in a future
version.
CLM needs to better represent the land cover after harvest to include the influence of
weeds and litter on the carbon balance. Although CLM properly simulated the seasonal
evolution of NEE, the NEE RMSE at US-ARM and US-CRT (2-3 umol/m2/s) is higher
than the Lund-Potsdam-Jena managed Land model (LPJ-ml) simulation (Bondeau et al.,
2007) at the US-PON site (1.09 umol/m2/s), which is largely due to incorrect simulation
of NEE after harvest. When winter wheat is not alive, CLM represents the land cover as
bare ground so GPP is zero but heterotrophic respiration from litter and soil organic
matter is still large, which resulted in a carbon source after harvest (positive NEE). This
is not true for the US-ARM site, where we observed weed growth after harvest and
positive NEE (Raz-Yaseef et al., 2015). This vegetation cover after harvest resulted in a
near zero NEE at US-ARM or negative NEE at US-CRT site. Appropriate simulation of
the post-harvest land cover is critical for better representing the role of agriculture in
global carbon fluxes.
CLM needs to further increase the influence of crops and vegetation on the surface
energy balance and latent heat flux (LE) in particular. The LE simulation in CLM has a
$R^2$ range from 0.62 to 0.97 across the four sites, which is better than other model
simulations at the same sites. For example, Arora et al., (2003) simulated LE RMSE 22.0
W/m$^2$ at US-PON from March-May in 1997 using their coupled land surface and
terrestrial ecosystem model (CLASS-Twoleaf model), and we simulated LE RMSE 10.55
W/m$^2$ at the same site from March-May averaged for 1998-1999. But our LE response to
the improved LAI was not as strong as we expected. Williams and Torn (2015) showed
that vegetation has stronger controls on surface heat flux partitioning than soil moisture at
the US-ARM site, where LAI explained 53% of the variation in evaporative fraction
(EF=LE/(LE+H)), while soil moisture only explained 11% of EF variation. For our six
winter wheat years (Williams and Torn (2015) used 8 years that included other cover
types), we found similar patterns in the US-ARM observations. LAI explained 40% of EF
variation while soil moisture only explained 7% (not shown). However, EF in CLMWHE
and CLMBASE was not as well predicted by LAI, which only explained 5% and 1%,
respectively, of variation in EF. In CLM, vegetation affects LE through leaf transpiration,
and LE in vegetated grid cells has three components: soil evaporation, wet leaf
evaporation, and dry leaf transpiration (Lawrence et al., 2007). The wet leaf evaporation
is the smallest and overall LE depends on the tradeoff between soil evaporation and leaf
transpiration. Soil evaporation is dominant when LAI is small, and leaf transpiration is
dominant when LAI is higher. Using the US-ARM site as an example, in CLMBASE, the
leaf transpiration is very small due to low LAI but soil evaporation is very large, which is
opposite in CLMWHE (Figure 4 a and b). Such a tradeoff is why the large increase in
LAI in CLMWHE only increased overall LE a small amount compared to CLMBASE.
We found that although the new soil evaporation parameterization (Swenson and
Lawrence, 2014) in a later version of CLM reduced soil evaporation and improved the
summer and fall LE simulation (Figure 4), it also reduced spring soil evaporation (Figure
4b) and induced an even lower spring LE. If we assume this reduction in soil evaporation
is reasonable, then further improvement of the LE simulation needs to be focused on
increasing the leaf transpiration and correcting the inconsistent peak time between leaf
transpiration and LAI.
CLMWHE tends to underestimate the winter wheat yield but the yield RMSE is
comparable to other literature values. The averaged yield RMSE across the four sites is
1.96 ton/ha, which was within the range of other winter wheat models yield RMSE (1.41-
2.15 ton/ha) reported by (Palosuo et al., 2011), although the simulation sites and years are
different. The low simulated yield may be due to the insufficient calibrations. Table 4
listed the key crop growth parameters used in CLMWHE. We calibrated these parameters
at the US-ARM site, and applied the same values everywhere, which is a common
approach in land surface model development. However, the US-ARM site represents a
relatively low yield compared to the U.S. national average. This likely contributed to
underestimated yields at sites or in regions with historically greater yields, such as at US-
CRT and CAF-CT, and in the Southeastern and Northwest US. The current modeling
framework of CLM does not facilitate the substantial calibration required to more
accurately capture the full range of observed winter wheat yields. As a gridded global
crop model, gridded parameters (e.g., maximum maturity days, leaf emerge and grain fill
threshold, and background litter fall factor) that allow for spatial variation in the key
parameters should be considered in future versions of the model. Alternately, for
parameters with spatial structure linked to environmental variation, parameters could
vary with climate or soil conditions.
We investigated the causes of the low yield in 2007 at the US-ARM site. The
observational yield data in Figure 4 is from the county level USDA yield estimate, which
is very similar (RMSE=0.11 ton/ha) to the US-ARM site-observed yield. Both the site-
observed yield and USDA county-level yield showed the lowest values in 2007 (1.35
ton/ha), so the low yield in 2007 is not specific to the field represented by the US-ARM
site.  The field notes indicate that only part of the wheat field was harvested in early July
of 2007, while the remainder of the field was not harvested due to wheat sprouting in the
head. Pre-harvest sprouting reduces the quality (and price) of the grain, and can occur
when the crop is exposed to prolonged heavy rain. We examined the precipitation,
temperature, and wind speed during May and June across the eight years and found that
in 2007 there was double the mean precipitation in June (108.2% higher than the eight-
year June average). Such large amounts of precipitation may have caused the low
observed yield. Assuming that the low yield was strongly linked to the high rainfall, the
implication is that the winter wheat crop model needs to include more types of
environmental damage to fully simulate interannual variation in yields.
Our new winter wheat model improved the LAI and yield simulation compared to the
original winter wheat model except at CAF-CT site due to 1) drier soil conditions during
the grain fill phase and 2) the adjusted grain carbon allocation coefficient in CLMWHE.
CLMWHE started the grain fill phase during the end of May while CLMBASE started
the grain fill phase in the beginning of May. In mid-May, the higher LAI in CLMWHE
resulted 30% more LE than CLMBASE and dried the soil. The plant wetness factor
dropped from 0.98 on May 15 to 0.19 on May 28 in CLMWHE, but remained greater
than 0.89 through May in CLMBASE. The grain carbon allocation in CLMWHE is
strongly limited by soil water available to the plant, so grain carbon was much smaller in
CLMWHE than in CLMBASE. The larger LAI also increased LE at the other three sites
relative to the baseline simulations, but did not result in long-term water stress due to
sufficient precipitation during the rainy season. The CAF-CT site has ten times less
precipitation than the other three sites in May. The observed LE at CAF-CT site is much
higher than the simulation given the same precipitation, suggesting the plant wetness
factor in the model is too sensitive to low precipitation.
Some of our modeling approaches need further improvements to the processes supported
by new observations. We developed hypothetical (empirically-based) frost damage
functions that account for both small and frequent damage early in the growing season,
and severe damage in winter and spring. Such a hypothetical approach is not uncommon
in crop modeling when lacking observations at a process-level. For example, CERES-
Wheat (Ritchie and Otter, 1985) developed a hypothetical leaf senescence scheme during
cold temperature that monitored a cold hardening index
(http://nowlin.css.msu.edu/wheat_book/CHAPTER3.html ). We tested the CERES-Wheat
leaf senescence scheme in CLM and found it produced too much reduction in LAI. This
finding motivated our approach based on recently developed frost tolerance indicators.
The magnitude of the leaf carbon reductions and how such reductions are linked to frost
damage requires more observations, such as high frequency aboveground and
belowground biomass measurements. Furthermore, the linear yield regressions showed
that the yields in CLM depend too much on growing degree days, a sensitivity that is not
reflected in observations. In CLM, growing degree days not only determine crop
phenology but are also involved in calculation of the carbon allocation coefficients (Table
3). Exploring other possible factors that control phenology and carbon allocation may
improve crop simulation in CLM. Meanwhile, soil moisture, especially the deeper soil
moisture, explains a large amount of the yield variation in both observations and the
simulations. Fixing the current biases in soil hydrology and reducing interannual
variability in the simulated soil moisture will benefit the yield simulation.
In summary, we found that our new winter wheat model in CLM better captured the
monthly variation of leaf area index and improved the latent heat flux and net ecosystem
exchange simulation in spring. Our model correctly simulated the interannual variation in
yield at the US-ARM site, but the crop growth calibration at the US-ARM site introduced
a low-yield bias that produced underestimates of the yield in high-yield sites (US-CRT
and CAF-CT) and regions (Northwestern and Southeastern US). Our analysis indicates
that while this model of winter wheat represents a substantial step forward in simulating
the processes that influence winter wheat growth and yield, further refinements would be
helpful to capture the impacts of environmental stress on energy partitioning, carbon
fluxes and yield, and would improve simulations of regional variation.
Code Availability
The winter wheat code in CLM4.5 can be requested from Yaqiong Lu
(yaqiong@ucar.edu). It will be available in the next released version of Community Land
Model (version 5) for public access.

Acknowledgements
This material is based upon work supported by the U.S. Department of Energy, Office of
Science, Office of Biological and Environmental Research, Atmospheric System
Research, under contract number DE-AC02-05CH11231. Funding for the US-ARM
AmeriFlux site was provided by the U.S. Department of Energy's Office of Science. This
research used resources of the National Energy Research Scientific Computing Center, a
DOE Office of Science User Facility supported by the Office of Science of the U.S.
Department of Energy under Contract No. DE-AC02-05CH11231. We acknowledge the
following additional AmeriFlux sites for their data records: US-ARM, US-PON, US-

CRT. In addition, funding for AmeriFlux data resources was provided by the U.S. Department of Energy's Office of Science. We also thank Sarah Waldo and Jinshu Chi at Washington State University for sharing the CAF-CT site data, and thank AgMIP-Wheat project for sharing the ABLE, NDMA, NESA, KSMA, TXLU site data.

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
