# Peer review of "Representing winter wheat in the Community Land Model (version 4.5)"

_Geoscientific Model Development, 2016_

## Referee Comment (RC1) · Anonymous Referee #1 · 27 Nov 2016

Review for "Development of a winter wheat model in the Community Land Model (version 4.5)" by Lu et al. Summary: Lu et al. updated the winter wheat model in the Community Land Model (CLM) to better simulate wheat growth and grain production, including schemes to represent vernalization and frost tolerance. They also validated the model with three observation data and then applied the updated the model on regional scale. The topic is interesting, but I have a few questions about the method and some comments as listed below. Comments: 1. The title "Development of a winter wheat model ......" indicates a new wheat model was implemented in the CLM. As far as I understood, they just updated vernalization and frost schemes. 2. In abstract, they claim that they calibrated the three key parameters. But I did not see how they did the calibration and which data did they use to calibrate. I have no idea about what is the difference between calibrated model and the model with default setting? 3. The implemented schemes (vernalization and frost tolerance) are the key contribution of this paper. However why they decided to choose the algorithm presented in this study is not clear for me as a number of the algorithm exists. Ideally, it would be great that they can validate these two schemes specifically instead of only validating the model in general. 4. In terms of frost damage, it is a very good point of this paper as climate extreme events are more frequent. However, they did not really show the improvement of the new frost scheme in predicting the frost events and quantify the damage of the frost. It would be promote the paper into a higher level if authors can validate and quantify the frost damage in plot scale, especially quantify the damage in region scale simulation. 5. The manuscript could be better if authors can tight the introduction. From results and discussion, I think the updating and validating of the model to estimate grain production are focus this paper, but they discussed a lot of the importance of carbon emission, energy and water exchange etc. Adding introduction about the importance growth stages would be very helpful. This may link to your decision why you want to focus on updating vernalization and frost schemes and ignore other processes. 6. In the introduction, they discuss a couple of the wheat models from plot to region scale. But what are the issues or challenges of these models did not mention. How they address these issues is not clear. 7. It is also important to note that they did not really validate the model with yield data even they use the model to simulate the wheat yield on regional scale. It would be much more convincible if the model validated on plot scale with grain production data. 8. My last comment is that updating the CLM-wheat model is important, but not very new topic as this kind of job has been done for some land surface models such as JULES, ORCHEDEE and BIO-BGC model. In short, this manuscript potentially is publishable, but it needs a number of modifications. Hence I would suggest a major revision.
* * *

---

## Referee Comment (RC2) · Anonymous Referee #2 · 15 Dec 2016

Manuscript GMD-2016-245

Authors:     Yaqiong Lu, I.N. Williams, J.E. Bagley, M.S. Torn, L.M. Kueppers
Title:       Development of a winter wheat model in the Community Land Model
             (version 4.5)

General remarks:

The paper contributes to on an interesting and important question: How to improve wheat growth models for a better prediction of future global grain production? For this purpose the paper addresses in particular the question how to better simulate leaf area

growth of winter wheat. It reports on the update of an existing crop growth model to the actual state of the art of describing vernalization, frost tolerance and damage, and on an improved parametrisation of carbon allocation to leaf and stem. The improved model is tested considering data sets from four different sites and up to six different winter seasons for each site across the USA representing different climatic conditions.

Generally I have difficulties to understand the focus of the paper: Is it the test of the proposed model update by comparison with data on frost impact on winter wheat growth and carbon allocation at the plot scale (fig.2-4) , or is it a general comparison with simulation results between the old (CLMBASE)and the new (CLMWHE) model at the US wide scale(fig.6-7) to show improvements?:

In the abstract the authors promise a validation of the updated model, but in my view the data sets for testing the simulation of leaf development (as dependent on vernalization and frost) are not sufficient to evaluate the presented algorithms of process descriptions (eq. 1-12): Although figures (fig.2) show for the sites US-ARM and US-PON improved LAI simulations, it is difficult to understand why this is the case: It should be carefully analysed if the improvement is based on a now more adequate description of frost damage/tolerance and/or on a more accurate description/parametrisation of carbon allocation to the leaves. Also, it is not clear to me if the test is performed using independent data sets which are not used for model parametrisation (in respect to both LAI and yield data): It would be helpful for the reader if the authors clearly indicate which data are used for model parametrisation and which data for model testing.

In summary, the paper could be considerably improved, if the proposed model update (eq. 1-12) is tested by data that are directly related to the underlying model mechanisms, i.e. by data on vernalization, frost damage and frost tolerance and by data on carbon allocation to leaves, stems and roots. Since both the model update and the allocation scheme are not new and the coupling of crop growth models to land surface models such as CLM has been successfully achieved already by different research groups, the paper would gain scientific value by a more thorough test.

[Figure]

In conclusion, the paper needs a total revision to ensure that the objective is met, i.e. of assessing the value of a more detailed description to improve model application at field and regional scale studies. In its present form the paper should not be published (see also special remarks).
Special remarks:

p=page,   l=line

| | | | |
|---|---|---|---|
| p=3 | l=93 | : | Is the model you use for update the same as the AgroIBIS model? |
| | | | If not, which are the differences? |
| | l=106 | : | Please insert the year: ... Levis et al. (2012)... |
| | l=107 | : | "has never been validated": at which scale? |
| | | | was the the AgroIBIS never tested? |
| | l=136 | : | Williams and Torn (2015) is not in the list of references. |
| p=4 | l=142 | : | Are there data on possible frost damage included? |
| p=5 | l=165-166 | : | Are these algorithms used in parallel? |
| | l=172 | : | Why are exactly these depths considered? |
| p=7 | l=242 | : | ... $LT_{50}$... |
| | l=274 | : | ... $LT_{50}$... |
| p=8 | l=310 | : | ... be likely to suffer as much from ... |
| p=9 | l=336 | : | ... the approach resulted in a rapid decline of ... |
| p=10 | l=361 | : | Was this performed for both models in the same way? |
| | l=278 | : | How long is a longer time? |
| p=11 | l=404 | : | Was this really achieved by better simulation of vernalization |
| | | | and frost damage? |
| p=12 | fig.2 | : | Why are there no error bars for CLMBASE? |
| | | | Was the lower observed LAI due to frost damage? |
| | | | Did you simulate some damage? |
| p=13 | l=458-461 | : | Legend of fig.3: "CLMCWHE" is this another model |
| | | | or is this a typo? |
| p=14 | l=498-500 | : | But now you may underestimate spring evaporation |
| | | | as strongly as you overestimated in later months. |
| | | | Please discuss in the discussion section. |

Moreover, observed times of LAI peaks are the same as LE peaks, at both sites, where you have observations!
Therfore, I don't understand l=498-500.

| p=15 | l=515 | : | bu/ac? Please use SI units throughout the paper. |
| p=17 | fig.6 | : | Could you compare with CLMBASE simulation? Is there an improvement compared to CLMBASE simulations and compared to US yield statistics? |
| p=18 | l=581-582 | : | Is this really an improvement, if you don't know that you simulate the processes correctly? In my view you should have compared with observed data to justify such an statement. |
| p=19 | l=617 | : | (Williams and Torn ... Please insert bracket and year of publication. |
|      | l=636 | : | There is overestimation for US-ARM and US-PON, see fig.5 Please correct your sentence: tends to underestimate US national average. |

---

## Author Response (AR1)

Dear Christoph,

Below please find responses to each review comment that are slightly revised from the last response letter, including details of how we modified the manuscript to address reviewer requests. Major changes include revisions to the introduction to more clearly articulate the aims of our work and it's expected value to the scientific community, and additional site validations to help with interpreting model-data comparisons.

We look forward to a favorable decision on our manuscript.

Sincerely,
Yaqiong Lu

**Review 1**

Review for "Development of a winter wheat model in the Community Land Model (version 4.5)" by Lu et al. Summary: Lu et al. updated the winter wheat model in the Community Land Model (CLM) to better simulate wheat growth and grain production, including schemes to represent vernalization and frost tolerance. They also validated the model with three observation data and then applied the updated the model on regional scale. The topic is interesting, but I have a few questions about the method and some comments as listed below.

Comments:
1. The title "Development of a winter wheat model…" indicates a new wheat model was implemented in the CLM. As far as I understood, they just updated vernalization and frost schemes.

Previously, when the crop models in CLM were developed, a wheat scheme was implemented, however this scheme did not reasonable represent winter crops (crops planted in fall for late spring harvest). Our work created this capability building off of the pre-existing wheat model code. To acknowledge the earlier wheat model development, we have modified the title to "Representing winter wheat in the Community Land Model (version 4.5)"

2. In abstract, they claim that they calibrated the three key parameters. But I did not see how they did the calibration and which data they used to calibrate. What is the difference between the calibrated model and the model with the default setting?

We calibrated the phenology and carbon allocation parameters to achieve good representation of US-ARM site measurements of LAI and yield, and tested the calibrated model using independent data sets from three other eddy flux sites (US-Pon, US-CRT, CAF-CT) and five additional winter wheat sites which have data on LAI, biomass, and grain weight.

To clarify, we added two sub-sections to the revised methods section (calibration data and validation data), to more clearly state the data used for calibration and validation. We showed in Table 4 the values of the parameters for the calibrated model (CLMWHE) and the model with

default setting (CLMBASE). Only three of the parameters are different between CLMWHE and CLMBASE and we indicated these modified parameters with superscript *.

We added a paragraph in the revised manuscript at line 381-386 to clarify how we calibrated the model:

"After the above modification of carbon allocation and addition of the new vernalization and frost damage processes, we calibrated three parameter values (indicated with * in Table 4) in the US-ARM simulation. We adjusted minimum planting temperature and maximum days for growing to better match the US-ARM site planting and harvest date, and adjusted the initial leaf carbon allocation coefficient to better match the US-ARM LAI and yield."

3. The implemented schemes (vernalization and frost tolerance) are the key contribution of this paper. However why they decided to choose the algorithm presented in this study is not clear for me as a number of the algorithm exists. Ideally, it would be great that they can validate these two schemes specifically instead of only validating the model in general.

We chose these algorithms because they were developed as generalized standalone models that could be easily added into CLM. Vernalization and frost tolerance (Equation 1 to 10) are based on previously developed models that have already been calibrated and tested against growth chamber or site measurements. We understand that in model coupling, calibration of some parameters may be necessary even if those parameters were previously calibrated on their own. For example, when Levis et al., (2012) coupled the crop growth module from AgroIBIS into CLM, they calibrated the phenology and carbon allocation parameters because other processes (e.g., soil hydrology, radiation, photosynthesis) in CLM were quite different from AgroIBIS. These different processes resulted in an unexpected crop growth when using the same parameters as in AgroIBIS. But this is not the case for our coupling of vernalization and frost tolerance processes into CLM. These two processes mainly rely on the air temperature and snow, which are largely dependent on the atmospheric forcing for the offline simulation. So the previous calibrations on the empirical functions should be valid for the coupled model.

4. In terms of frost damage, it is a very good point of this paper as climate extreme events are more frequent. However, they did not really show the improvement of the new frost scheme in predicting the frost events and quantify the damage of the frost. It would promote the paper into a higher level if authors can validate and quantify the frost damage in plot scale, especially quantify the damage in region scale simulation.

For frost damage, we don't have specific measurements of tissue biomass at the four flux tower sites. We did a literature search on how other winter wheat models validated their frost damage algorithms, but didn't find any validations of this specific process. We've contacted several agronomic groups for field experiments that specifically measured frost damage, but had no luck with responses. But we did get five winter wheat sites reformatted and used by the AgMIP-Wheat project. These sites have detailed crop growth measurements (e.g., leaf area index, leaf, stem, and grain weight). We added the validation at the five sites for leaf area index, leaf, stem, and grain weight to show the performance of the model (line 448-476).

Following the reviewer's suggestion, we quantified the effect of frost damage on U.S. winter wheat LAI and yield through a sensitivity analysis. We compared two US domain simulations with the updated winter wheat model: one with the active frost damage function, and the other with the frost damage function turned off. These supporting results were added to the revised manuscript, along with the following text at line 613-623:

"We quantified frost damage impacts on LAI and yield in the US domain through CLMWHE simulations with and without the frost damage function. Frost damage resulted in lower LAI and yield, with spatial variation across the U.S (Figure 8). For the domain average, frost damage reduced LAI by 27% (or 1.69 m2/m2) and reduced yield by 28% (or 0.5 ton/ha). The greatest reduction (>45%) in LAI occurred in Texas and the southeastern US, which was due to insufficient hardening, producing a high LT50 and low survival rate. LAI in the cold northern US regions had less impact (<15%) from frost damage. The cold damage indirectly affects yield through reduced photosynthesis with lower LAI, but photosynthesis and yield changes were not always geographically consistent with the LAI damage. For example, the northern Great Plains and Midwest had greater percentage reductions (>45%) in yield than reductions in LAI (< 15%)."

5. The manuscript could be better if authors can tighten the introduction. From results and discussion, I think the updating and validating of the model to estimate grain production are focus this paper, but they discussed a lot of the importance of carbon emission, energy and water exchange etc. Adding introduction about the importance growth stages would be very helpful. This may link to your decision why you want to focus on updating vernalization and frost schemes and ignore other processes.

We had at least two reasons for representing winter wheat in CLM, (1) the importance of winter wheat as a land cover type (particularly in the Southern Great Plains of the U.S.), and its seasonally unique control on carbon, energy and water exchange with the atmosphere and climate; and (2) the distinct importance of winter wheat (vs spring wheat) in contributing to regional and global crop productivity and yield, and differences in its sensitivity to climate variability and extremes. We revised the introduction to better motivate the importance of representing winter wheat energy, water and carbon exchange, as well as yield and the sensitivity of yield to vernalization and frost at line 45-61:

"Winter wheat, which is planted in fall and harvested in early summer, has a different growth cycle and responds to environmental stresses differently from summer crops. Winter wheat may suffer less from summer drought but is subject to winter damage due to exposure to low temperatures and frequent freeze-thaw cycles (Vico et al., 2014). There are two important over-winter survival mechanisms for winter wheat: vernalization and cold tolerance. Vernalization is the process whereby winter wheat is exposed to a period of non-lethal low temperature required to fully enter the flowering stage and to produce grain in spring. Additionally, winter wheat acclimates to low temperature giving it the capability to survive cold temperatures. Both of these processes – vernalization and cold tolerance - are cumulative processes and have similar optimum temperature ranges. When the temperature is outside of the optimum range, the processes can be stopped, reversed, and restarted (Fowler et al., 1999). Damage can occur when temperatures are lower than the accumulated cold tolerance (reviewed by Barlow et al., 2015). Cold-induced damage has been observed to persist through the remainder of the growing season,

and its impact on yield is greater than on growth. Effectively representing these processes in crop models could improve understanding of the uncertainty in the future crop yield projections."

6. In the introduction, they discuss a couple of the wheat models from plot to region scale. But what are the issues or challenges of these models? How they address these issues is not clear.

The primary challenge is in representing the unique phenology of winter wheat when coupled to a biochemical (as opposed to light-use efficiency) model of photosynthesis. This includes representing vernalization and cold tolerance. A secondary challenge is representing winter wheat productivity and yield at regional and global scales, as opposed to optimizing a model for local conditions.

Our revised introduction includes additional background on how these challenges have been addressed in crop models, and how they can be implemented in the land-surface models used in Earth system models at line 86-108.

"In contrast to their strength in representing crop growth processes, these models have simplified treatment of important upstream processes that affect crop growth. For example, the photosynthesis scheme is a linear function of intercepted photosynthetically active radiation (PAR), PAR itself is simplified as a constant fraction of incoming solar radiation, and radiation is not separated into direct and diffuse fractions. Further, these crop models were originally developed to simulate individual, as opposed to multiple crops, making multi-crop simulations at regional and global scales difficult.

To incorporate more physical processes and to simulate crop growth at regional or global scales, some agronomic crop growth models were incorporated into agro-ecosystem models. For example, CERES maize and wheat growth were added into the Decision Support System for Agrotechnology Transfer Model (DSSAT) (Jones et al., 2003). A substantial modified version of CERES Wheat (Keating et al., 2001) also has been added into the Agricultural Production Systems Simulator (APSIM) Model (Keating et al., 2003). As the effects of vegetation on the atmospheric boundary layer have been increasingly appreciated, some land surface models started to also incorporate crop growth models to not only simulate crop yield, but also to simulate crop growth effects on surface carbon, water, and energy fluxes. For example, the SUCROS crop growth model was incorporated to JULES (Van den Hoof et al., 2011) and the STIC crop growth model was incorporated to ORCHIDEE (Wu et al., 2016). In the recent Agricultural Model Intercomparison and Improvement Project (AgMIP), these agro-ecosystem models and land surface models were categorized as Global Gridded Crop Models (GGCM)."

7. It is also important to note that they did not really validate the model with yield data even they use the model to simulate the wheat yield on regional scale. It would be much more convincible if the model validated on plot scale with grain production data.

The reviewer must have overlooked the yield validation. For the flux tower sites, we had measured yield at the US-ARM site, but did not have yield information at the other three sites. Therefore, we compared modeled yield to the county yield data, which we found was consistent with the site-based measurements at US-ARM. We updated figure 5 to show the ARM site yield

observation. To further address the comment, we evaluated the model against independent data from an additional five winter wheat sites, to validate prediction of LAI, leaf, stem, and grain weight. And the results showed our model could well represent the temporal and spatial variation for winter wheat growth at these new sites.

8. My last comment is that updating the CLM-wheat model is important, but not very new topic as this kind of job has been done for some land surface models such as JULES, ORCHEDEE and BIO-BGC model. In short, this manuscript potentially is publishable, but it needs a number of modifications. Hence I would suggest a major revision.

CLM is one of the most widely used land surface models. The biogeochemical process in CLM3.5 were initially based on BIOM-BGC and then largely improved in the later version. CLM, JULES, and ORCHIDEE are land surface components in different Earth System models (CESM, Met Office Unified Model, IPSL-CM5). They all have sub-grid cell structures that represent 12 PFTs in ORCHIDEE, 5 PFTs in JULES, and 16 PFTs in CLM. They all started to incorporate crop growth models to improve the simulation of cropland in recent years. For CLM, the lack of a valid winter wheat model constrained its utility for studying interactions among climate, land use and agriculture at regional to global scales. We further developed and validated the winter wheat model in CLM to help the CLM community better understand the effects of crop growth on water and energy exchange, and to contribute more fully to international projects concerned with effects of climate variability and change on agriculture, such as AgMIP.

**Review 2**
The paper contributes to an interesting and important question: How to improve wheat growth models for a better prediction of future global grain production? For this purpose the paper addresses in particular the question how to better simulate leaf area growth of winter wheat. It reports on the update of an existing crop growth model to the actual state of the art of describing vernalization, frost tolerance and damage, and on an improved parameterization of carbon allocation to leaf and stem. The improved model is tested considering data sets from four different sites and up to six different winter seasons for each site across the USA representing different climatic conditions.

Generally I have difficulties to understand the focus of the paper: Is it the test of the proposed model update by comparison with data on frost impact on winter wheat growth and carbon allocation at the plot scale (fig.2-4), or is it a general comparison with simulation results between the old (CLMBASE)and the new (CLMWHE) model at the US wide scale(fig.6-7) to show improvements?

The focus of the paper is to add representation of winter wheat into CLM. We compared the results of our new code (CLMWHE) to the baseline model simulations (CLMBASE) to show the improvements over the original model, which did not include valid winter wheat and therefore had biases in land-atmosphere exchange of energy and water. We also compared the new code to observations to show the strengths and limitations of the model. We revised the introduction to reflect our dual aims of representing the unique effects of winter wheat on land-atmosphere energy and water fluxes (and therefore climate) and the productivity and yield of winter wheat as an important grain crop, including it's vulnerability to frost damage.

In the abstract the authors promise a validation of the updated model, but in my view the data sets for testing the simulation of leaf development (as dependent on vernalization and frost) are not sufficient to evaluate the presented algorithms of process descriptions (eq. 1-12): Although figures (fig.2) show for the sites US-ARM and US- PON improved LAI simulations, it is difficult to understand why this is the case: It should be carefully analysed if the improvement is based on a now more adequate description of frost damage/tolerance and/or on a more accurate description/parametrisation of carbon allocation to the leaves.

As noted above, the vernalization and frost tolerance parameterizations were previously calibrated and tested and were not further calibrated or tested here. (see our detailed response to Reviewer 1 Comment 3 and 4 above)

We have conducted an additional set of simulations to isolate, for monthly LAI, the effects of each change. However, vernalization and frost damage code interact with each other in nonlinear ways. And the parameters were calibrated after adding the vernalization and frost damage schemes, so it did not make sense to use them separately.

With the additional winter wheat site validation of leaf, stem, and grain weight, we found vernalization played the most important role in simulation of the crop growth seasonal cycles due to its controls on the growing degree days and phenology shifts, and the cold damage also played a role in simulation of a reasonable winter time LAI and leaf weight, but did not affect the phenology. We added these new results at line 448-470:

"The modified winter wheat model (CLMWHE) showed a better seasonal growth cycle than the original model (CLMBASE) (Figure 2). In the CLMBASE simulation, the vernalization factor is too high even at the beginning of the growing season (Figure S3). Without the reduction on the growing degree days from the vernalization function, winter wheat LAI and leaf weight reached peaks in December and then declined due to the onset of the grain fill stage. The long grain fill stage (December – May) in CLMBASE did not produce a sufficiently high grain mass because of the low rate of photosynthesis with the low LAI. In the CLMWHE simulation, LAI and leaf weight reached peaks in April, and stem and grain weight reached peaks in May, which are similar to the site observations. The improvements in the seasonal variation are mainly due to the updated vernalization, which produced a reasonable vernalization period about two-three months, reduced the growing degree days and extended the leaf emergence stage. The cold damage scheme also played a role in reasonable simulation of winter LAI and leaf weight. For example, at KSMA-1985, cold damage reduced the LAI and leaf weight in fall yielding a better match to the winter measurement (at DOY=320).

The updated winter wheat model captured the grain weight temporal and spatial variations, and RMSE and the index of agreement are better in CLMWHE than CLMBASE for seven site-years. CLMWHE showed higher grain weight in 1986 than 1985 at TXLU and NESA, as did the observations, because 1986 was a wetter year for both TXLU (8% higher annual precipitation than 1985) and NESA (84% higher). In 1986, CLMWHE showed more grain weight in NESA and NDMA than TXLU and ABLE, as in the observations."

Also, it is not clear to me if the test is performed using independent data sets which are not used for model parametrisation (in respect to both LAI and yield data): It would be helpful for the

reader if the authors clearly indicate which data are used for model parametrisation and which data for model testing.

In the revised method section, we split the site description into calibration data and validation data sections. We more clearly stated what are the calibration data and what are the validation data (see line 152-184).

In summary, the paper could be considerably improved, if the proposed model update (eq. 1-12) is tested by data that are directly related to the underlying model mechanisms, i.e. by data on vernalization, frost damage and frost tolerance and by data on carbon allocation to leaves, stems and roots. Since both the model update and the allocation scheme are not new and the coupling of crop growth models to land surface models such as CLM has been successfully achieved already by different research groups, the paper would gain scientific value by a more thorough test.

As noted above, we adopted algorithms for vernalization and frost damage that were already carefully developed and validated with data that we don't have access to. We extended our validation to leaf, stem, grain weight at an additional five winter wheat sites in the revised manuscript.

In conclusion, the paper needs a total revision to ensure that the objective is met, i.e. of assessing the value of a more detailed description to improve model application at field and regional scale studies. In its present form the paper should not be published (see also special remarks).

Prior to our work, CLM did not have a valid and tested winter wheat model. Our contribution to the literature and to the modeling community is to describe our revised and tested representation of winter wheat in CLM so that the community can understand and use the model for their scientific objectives. Revisions to the introduction will make this more explicit.

Special remarks:
p=page, l=line
p=3 l=93 : Is the model you use for update the same as the AgroIBIS model?
If not, which are the differences?

CLM crop growth only adopted the crop phenology and carbon allocation from Agro-IBIS, and did not include other processes in Agro-IBIS (e.g., hydrology, belowground nitrogen dynamics, radiation). The parameters in crop phenology and carbon allocation have different values from Agro-IBIS with calibrations described in Levis et al., 2012.

l=106 : Please insert the year: ... Levis et al. (2012)...

Inserted year.

l=107 : "has never been validated": at which scale? was the the AgroIBIS never tested?

Prior to our work, CLM did not represent winter wheat - all wheat was spring wheat. To our knowledge, AgroIBIS was never validated at winter wheat sites, and even if AgroIBIS had been

validated for winter wheat, winter wheat code and parameterizations were not implemented in CLM. This sentence will be revised to avoid confusion.

l=136 : Williams and Torn (2015) is not in the list of references.

added

p=4 l=142 : Are there data on possible frost damage included?

No, the site has no specific measurements on frost damage.

p=5 l=165-166 : Are these algorithms used in parallel?

The algorithms could be used in serial or parallel. CLM can support single point simulations, regional simulations, and global simulations. The single point simulation is a serial job, while the regional and global simulations can be run in parallel, where the domain is divided into sub regions, and run on multiple processors.

l=172 : Why are exactly these depths considered?

These depths were considered the top soil layer that affects seed germination most.

p=7 l=242 : ... *LT50*...

Modified to Italics font.

l=274 : ... *LT50*...

Modified to Italics font.

p=8 l=310 : ... be likely to suffer as much from ...

Modified

p=9 l=336 : ... the approach resulted in a rapid decline of ...

Modified

p=10 l=361 : Was this performed for both models in the same way?

The spinup for CLMWHE and CLMBASE are quite similar. We used the same spin up from the new model (CLMWHE) as the initial condition for CLMBASE for consistency.

l=378 : How long is a longer time?

Hundreds years (see table S1).

p=11 l=404 : Was this really achieved by better simulation of vernalization
and frost damage?

Vernalization played the most important role in simulating the seasonal cycle of crop growth due
to its control on growing degree days and phenological stage. Cold damage also played a role in
simulation of a reasonable winter time LAI and leaf weight.

p=12 fig.2 : Why are there no error bars for CLMBASE? Was the lower observed LAI due to
frost damage? Did you simulate some damage?

US-ARM and US-PON have error bars (very small) for CLMBASE. US-CRT and CAF-CT only
have one year so there are no error bars for both CLMBASE and CLMWHE.

It is hard to say whether the lower observed LAI at US-PON is due to frost damage or not, we
did simulate slight damage (LAI reduction $< 0.3$ m$^2$/m$^2$).

p=13 l=458-461 : Legend of fig.3: "CLMCWHE" is this another model
or is this a typo?

We corrected the typo.

p=14 l=498-500 : But now you may underestimate spring evaporation
as strongly as you overestimated in later months.
Please discuss in the discussion section.

We added to the discussion at line 700-706:

"We found that although the new soil evaporation parameterization (Swenson and Lawrence,
2014) in a later version of CLM reduced soil evaporation and improved the summer and fall LE
simulation (Figure 4), it also reduced spring soil evaporation (Figure 4b) and induced an even
lower spring LE. If we assume this reduction in soil evaporation is reasonable, then further
improvement of the LE simulation needs to be focused on increasing the leaf transpiration and
correcting the inconsistent peak time between leaf transpiration and LAI."

Moreover, observed times of LAI peaks are the same as LE peaks,
at both sites, where you have observations!
Therfore, I don't understand l=498-500.

In the observation, LE peaks at the same time as LAI peaks in April. But in the CLMWHE
simulation, simulated LE peaks in May while simulated LAI peaks in April. The new soil
evaporation scheme still has not solved the inconsistent LE-LAI peak problem. Here we simply
stated that the LE peaks in May due to the fact that its dominant component, leaf transpiration,
peaks in May.

p=15 l=515 : bu/ac? Please use SI units throughout the paper.

We changed the yield unit from bu/ac to ton/ha.

p=17 fig.6 : Could you compare with CLMBASE simulation?
Is there an improvement compared to CLMBASE
simulations and compared to US yield statistics?

We added figure 6c to show the CLMBASE US simulation. CLMWHE (Figure 6b) showed a
better US yield estimation (RMSE reduced by 24%) than CLMBASE (Figure 6c).

p=18 l=581-582 : Is this really an improvement, if you don't know that you simulate the
processes correctly? In my view you should have compared with observed data to justify such a
statement.

As we mentioned earlier, there is no need to validate the vernalization and cold tolerance
functions independently because they've been previously validated. We could not fully validate
the frost damage function due to observation limitations. But using data from additional winter
wheat sites we were able to further validate leaf, stem, and grain weight. We found that
vernalization played the most important role in simulating the seasonal cycle of crop growth due
to its control on growing degree days and phenological stage. Cold damage also played a role in
simulation of a reasonable winter time LAI and leaf mass.

p=19 l=617 : (Williams and Torn ... Please insert bracket and year of publication.

Added.

l=636 : There is overestimation for US-ARM and US-PON, see fig.5
Please correct your sentence: tends to underestimate
US national average.

Corrected.

---

## Referee Report (RR1)

General comments:

I reviewed the manuscript "Representing winter wheat in the Community Land Model (version 4.5)" drafted by Lu *et al*. The main contribution of this paper is improving the winter wheat representation in CLM, by modifying the vernalization, frost damage, and carbon allocation scheme etc. In general, the model structure and function is explained clearly, but some concerns should be clarified further, including:

1. what is the nitrogen limitation effect on the winter wheat growth and grain yield? Do you consider it? If not, please expand this part a little bit more.
2. I notice that your model generally overestimate the LAI for all simulations on TXLU, KSMA, NESA, NDMA, ABLE, especially at the latter of growing season, but it simulate well at US-ARM. Can you explain a little bit more about it? Nitrogen? Or you do not have leaf senescence process in your model?
3. where do you get the key equations for improving the winter wheat representation? I did not see the exact literatures for most of those equations?

Please also see my specific comments below.

Specific comments

Line 19, is this module a new one? Or you just modify some specific processes on this module? If so, I suggested to change this sentence to one like " We modified xxx or adapted xxx "

Line 21, use the subscript

Line 28, add some numeric evidence, such as how much reduction in RMSE?
Line 30, to what extent does it underestimate winter wheat yield?

Line 54, literature?
Line 59, literature?

Line 173, there is no irrigation, right? I am not sure which sites do you finally use to validate your model, all or just some of them? You mentioned that there is nitrogen and irrigation experiment on these sites, but finally you select seven site-years rainfed plots. It is not clear.

Line 204, what is the threshold of the maximum daily increment?
Line 206, literature is needed.

Line 211, what is the planting depth for seeds?

Line 235-248, literature?

Line 252, you mentioned that the VF affects the grain filling with same extent to growth. But the VF is effective during leaf emergence to flowering. As far as I know the grain filling starts after flowering. How does it affect grain filling, by heading? Please clarify it.

Line 465, to what extent?

L505, generally, there is energy closure problem at EC observations, and do you figure out the problem in LE?

Line 586, do you compare your model simulation with observations from only rainfed regions or all winter wheat regions? I suggest to compare your model results with that from rainfed regions.

---

## Author Response (AR2)

General comments:

I reviewed the manuscript "Representing winter wheat in the Community Land Model (version 4.5)" drafted by Lu et al. The main contribution of this paper is improving the winter wheat representation in CLM, by modifying the vernalization, frost damage, and carbon allocation scheme etc. In general, the model structure and function is explained clearly, but some concerns should be clarified further, including:

1. what is the nitrogen limitation effect on the winter wheat growth and grain yield? Do you consider it? If not, please expand this part a little bit more.

We clarified this at line 434-444:

"We applied the nitrogen fertilization in all the simulations. CLM4.5 considered the nitrogen limitation through the down regulation of the potential photosynthesis based on the nitrogen demand and supply deficit, which was calculated by considering the complex below ground biogeochemical processes (e.g., nitrification, denitrification, leaching, soil organic matter decomposition). When nitrogen supply is less than the nitrogen demand, the potential photosynthesis will be reduced by the deficit factor. For the TXLU, KSMA, NESA, NDMA, and ABLE site simulations, we applied the observed nitrogen fertilization amount (10-20 gN/m2) at the same days as the observation. While for the other sites and the US simulations, we applied the default nitrogen fertilization during leaf emergence every year for an amount of 8gN/m2. With these nitrogen fertilization, there are no nitrogen limitation at all our simulations."

2. I notice that your model generally overestimate the LAI for all simulations on TXLU, KSMA, NESA, NDMA, ABLE, especially at the latter of growing season, but it simulate well at US-ARM. Can you explain a little bit more about it? Nitrogen? Or you do not have leaf senescence process in your model?

Such overestimation is due to the leaf senescence rate is too low. CLM considered the leaf senescence in the later growing season when crop enters the grain fill period. The US-ARM site actually showed a similar overestimation for the later growing season LAI. Such overestimation was averaged out for the monthly average plot (Figure 3a). An improvement plan is actually taken place to fix such problem. We pointed out such deficiency at line 475-477:

"Besides these improvements, we also observed an overestimation of LAI during the later growing season, which is due to the low leaf senescence rate during the grain fill period."

3. where do you get the key equations for improving the winter wheat representation? I did not see the exact literatures for most of those equations?

The equation 1 to 10 were directly adopted from literature without any modifications. Specifically, equation 1 to 3 are from equation 2-5 in Streck et al., 2003. Equation 4-8 are from equation 1-5 in Bergjor et al., 2008. Equation 9-10 are from equation 1-2 in Vico et al., 2014. While equation 11 to 12 are our own empirical frost damage functions.

Vico, G., Hurry, V., and Weih, M.: Snowed in for survival: Quantifying the risk of winter damage to overwintering field crops in northern temperate latitudes, Agr Forest Meteorol, 197, 65-75, 2014.
Streck, N. A., Weiss, A., and Baenziger, P. S.: A generalized vernalization response function for winter wheat, Agron J, 95, 155-159, 2003.
Bergjord, A. K., Bonesmo, H., and Skjelvag, A. O.: Modelling the course of frost tolerance in winter wheat I. Model development, Eur J Agron, 28, 321-330, 2008.

Please also see my specific comments below. Specific comments

Line 19, is this module a new one? Or you just modify some specific processes on this module? If so, I suggested to change this sentence to one like " We modified xxx or adapted xxx "

We changed the sentence to "We modified the winter wheat model.."

Line 21, use the subscript

Modified.

Line 28, add some numeric evidence, such as how much reduction in RMSE? Line 30, to what extent does it underestimate winter wheat yield?

We added the numeric evidence for the two statements.

"reduced latent heat flux and net ecosystem exchange RMSE by 41% and 35% during the spring growing season."

"historically greater yields by 35%."

Line 54, literature? Line 59, literature?

We added Chouard (1960) for vernalization process at line 54, and updated the literature at line 59.

Line 173, there is no irrigation, right? I am not sure which sites do you finally use to validate your model, all or just some of them? You mentioned that there is nitrogen and irrigation experiment on these sites, but finally you select seven site-years rainfed plots. It is not clear.

We validated to all the five sites at rainfed years only. We pointed the site-year at line 173-174:

"For our validations, we only validated to seven site-year rainfed plots, which are TXLU-1985&1986, KSMA-1985, NESA-1985&1986, NDMA-1986, and ABLE-1986."

Line 204, what is the threshold of the maximum daily increment? Line 206, literature is needed.

The maximum daily increment is 26 $^o$day. We added Levis et al., 2012 to line 206.

Line 211, what is the planting depth for seeds? Line 235-248, literature?

CLM don't simulate the exact planting depth for seeds. We use the top two-layer soil temperature as a general estimation for the soil temperature that might affect seed germination. The equations at line 235-248 are from Streck et al., 2003, which has been added at line 220.

Line 252, you mentioned that the VF affects the grain filling with same extent to growth. But the VF is effective during leaf emergence to flowering. As far as I know the grain filling starts after flowering. How does it affect grain filling, by heading? Please clarify it.

VF was calculated from leaf emergence to flowering, but will affect the whole growing season through its impact on growing degree days. If VF is less than 1 (not fully vernalized), then $GDD_{plant}$ and grain carbon allocation coefficient will be both low, which will extend the leaf emerge period and reduce dry matter allocation to grain yield.

Line 465, to what extent?

We added: "RMSE was reduced by 19% and index of agreement was increased by 45%."

L505, generally, there is energy closure problem at EC observations, and do you figure out the problem in LE?

The energy imbalance in EC system may resulted a lower observed LE. The ARM site has about 25% energy imbalance, but it will not affect our major conclusion: the simulated LAI peak not resulted a LE peak.

Line 586, do you compare your model simulation with observations from only rainfed regions or all winter wheat regions? I suggest to compare your model results with that from rainfed regions.

Yes, we only compared to USDA non-irrigated winter wheat yield.

---

## Author Response (AR3)

Dear Christoph,

Thank you very much for these detailed comments. We added the clarifications to address each comment.

Sincerely,
Yaqiong Lu thanks for the revisions. Before accepting it for publication, I feel that your answers to the reviewer comments should also be reflected by changes in the manuscript.
* You have adequately responded to general comment 3 on the origin of the equations, but when reading the manuscript this is still not entirely clear. Please add the information of your response also to the manuscript in adequate form "eq 1-3 are taken directly from Streck; eq 4-8 are from Bergjor, 9-10 are from Vico and 11-12 are our own empirical functions" or similar.

We added clarifications at:

Line 227 "(equation 1-3 were directly adopted from Streck et al., (2003))."
Line 266-268 "Here, equation 4-8 were from Bergjord et al., (2008) and equation 9-10 were from Vico et al., (2014), without any modifications."
Line 341-342 "Here we developed our own hypothetical two-stage frost damage parameterization (equation 11-12)"

* similarly, I would like to see that your answer to the energy balance closure problem (comment on L505) can be found in the manuscript as certainly also other readers will have that question

We added clarifications at:

Line 181-187 "One caveat of using the eddy flux observation is the energy balance closure problem (Foken, 2008; Wilson et al., 2002) due to the eddy covariance technique limitation or the errors in calculating energy fluxes terms. The energy closure ratio at the four eddy flux sites are 87% at US-ARM, 91% at US-PON, 70% at US-CRT, and 83% at CAF-CT during the period used in the comparison. We used these imbalanced energy fluxes as is and discussed their impacts on our results."

Line 561-565 "The lack of energy balance closure for the eddy flux measurements could affect the energy fluxes RMSE estimations but will not change the major conclusions here: CLMWHE showed improved spring LE simulations than CLMBASE, and the simulated LE peak was one month later than LAI peaks."

* and finally also the point that you are only comparing to data from USDA non-irrigated winter wheat yield observations is not clear in the manuscript but clearly should be included there.

We added clarifications at:

Line 190 "non-irrigated winter wheat yield data"
Line 191-192 "with the nearest county non-irrigated yield."
Line 615 "non-irrigated winter wheat yield"

[revised manuscript text omitted]